

# Carbon balance of a grazed savanna grassland ecosystem in South Africa

Matti Räsänen[1], Mika Aurela[2], Ville Vakkari[2], Johan P. Beukes[3], Pieter G. Van Zyl[3], Miroslav Josipovic[3], Andrew D. Venter[3], Kerneels Jaars[3], Stefan J. Siebert[3], Tuomas Laurila[2], Juha-Pekka Tuovinen[2], Janne Rinne[1,2,4,5], and Lauri Laakso[2,3]

[1]Department of Physics, University of Helsinki, Finland
[2]Finnish Meteorological Institute, Helsinki, Finland
[3]Unit for Environmental Sciences and Management, North-West University, South Africa
[4]Department of Geosciences and Geography, University of Helsinki, Finland
[5]Department of Physical Geography and Ecosystem Science, Lund University, Sweden

*Correspondence to*: Matti Räsänen (matti.rasanen@helsinki.fi)

**Abstract.** Tropical savannas and grasslands are estimated to contribute significantly to the total primary production of all terrestrial vegetation. Large parts of African savannas and grasslands are used for agriculture and cattle grazing, but the carbon flux data available from these areas is limited. This study explores carbon dioxide fluxes measured with the eddy covariance method for three years at a grazed savanna grassland in Welgegund, South Africa. The tree cover around the measurement site, grazed by cattle and sheep, was around 15%. The night-time respiration was not significantly dependent on either soil moisture or soil temperature on a weekly temporal scale, whereas on an annual time scale higher respiration rates were observed when soil temperatures were higher. The yearly carbon dioxide balances of the growing seasons 2010-11, 2011-12 and 2012-13 were -85, 67 and 139 g C m$^{-2}$ yr$^{-1}$, respectively. The yearly variation was largely determined by the changes in the early wet season fluxes (September to November) and in the mid-growing season fluxes (December to January). Early rainfall enhanced the respiratory capacity of the ecosystem throughout the year, whereas during the mid-growing season high rainfall resulted in high carbon uptake. This study underlines the difficulty in establishing a functional relation between the total ecosystem respiration and the environmental drivers in savanna ecosystems. Furthermore, the high inter-annual variation of carbon balance in savanna ecosystems is difficult to relate to environmental drivers.



# 1 Introduction

Savannas are highly dynamic ecosystems which cover about 40% of Africa (Scholes and Walker, 1993) and 20% of the global land area. The seasonal cycle in savanna ecosystems is generally characterized by alternating wet and dry seasons, during the latter of which wildfires can occur. There are large differences between savannas in terms of their tree cover, species composition and soil type. Furthermore, large parts of African savannas have been inhabited by humans throughout the evolution of our species and thus are modified by anthropogenic activities such as grazing and logging.

Overall, the African continent is estimated to be a small sink of atmospheric carbon, although the uncertainty of this estimate is high due to the lack of long-term measurements in many key ecosystems of the continent (Valentini et al., 2014). The tropical savannas and grasslands are estimated to account for 30% of the global primary production of all terrestrial vegetation (Grace et al., 2006). In addition, it has recently been shown that the inter-annual variability of the terrestrial carbon cycle is dominated by semi-arid ecosystems (Ahlström, 2015).

The main meteorological drivers of the carbon fluxes between the atmosphere and African savannas are precipitation, soil moisture and soil temperature. The maximum carbon assimilation rates in a range of different African ecosystems have been shown to be an exponential function of the mean annual rainfall (Merbold et al., 2009). While the total ecosystem respiration was observed to be exponentially dependent on the soil temperature at seasonal time scales in South African savanna (Kutsch et al., 2008), Archibald et al. (2009) did not consider the conventional exponential function as an appropriate representation of the temperature response. Instead, they found that a generalized Poisson function was a better descriptor of the effect of temperature on respiration, as it describes both the exponential increase of respiration with the temperature and its decrease at higher temperatures.

Net Ecosystem Exchange (NEE) has been determined by eddy covariance at various savanna sites (Ago et al., 2014; Archibald et al., 2009; Brümmer et al., 2008; Quansah et al., 2015; Tagesson et al., 2015; Veenendaal et al., 2004). Most of these sites are located inside national parks or nature reserves. However, large parts of African savannas are used for agriculture and grazing; for example in South Africa 80% of the land surface is taken up by farmlands (Kotze and Rose, 2015).

The yearly sum of NEE has been observed to range between -429 (sink) and +155 (source) g C m$^{-2}$ yr$^{-1}$ across eight sites in semi-arid African savannas (Archibald et al., 2009; Brümmer et al., 2008). Archibald et al. (2009) found that the main drivers of inter-annual variation in NEE are the amount of absorbed Photosynthetically Active Radiation (PAR), length of the growing season and the number of days in the year when moisture was available in the soil. However, the environmental drivers for the inter-annual variation in NEE are poorly understood.

To understand these human-influenced savanna ecosystems, we analysed three years of eddy covariance CO$_2$ flux data from a grazed semi-arid savanna in central southern Africa. The carbon balance of this ecosystem was determined for a three-year period and its response to the environmental drivers was analysed at diurnal, monthly and inter-annual time scales. The longer-term productivity at the measurement site was assessed using a remotely sensed Normalised Differential Vegetation



Index (NDVI) as proxy for Gross Primary Production (GPP). The main objective of this study is to quantify the carbon balance and its inter-annual variation in a grazed semi-arid savanna ecosystem and to find possible climatic drivers for this variation.

## 2 Materials and methods

### 2.1 Site description

The Welgegund atmospheric measurement station (26°34'10"S, 26°56'21"E, 1480 m a.s.l.; www.welgegund.org) in South Africa has been measuring atmospheric aerosols and trace gases since May 2010. This site is located on a savanna grassland plain which is grazed by cattle and sheep. The monthly mean temperature and precipitation at a nearby weather station during 1998-2014 are shown in Figure 1. In general, the rainy season lasts from October to April, coinciding with the highest temperatures, but there can be substantial amount of rain as early as in September. This is followed by the dry and cooler season from May to September. The mean annual rainfall was 540 mm with a standard deviation of 112 mm between 1998 and 2014. During this period, on average, 93% of the yearly rainfall occurred between October and April. The main wind direction was from the north-west during daytime and the north-east during night-time.

The area around the eddy covariance measurement tower is dominated by perennial $C_4$ grass species (Table 1). The dominant grass species are *Eragrostis trichophora*, *Panicum maximum* and *Setaria sphacelata*. There is also a considerable amount of forbs, of which the dominant species are *Dicoma tomentosa*, *Hermannia depressa*, *Pentzia globosa* and *Walafrida densiflora*. This grassland type is referred to as a thornveld. It has a tree cover above 15% and an average tree height of 2.5 m. The common tree species are *Vachellia erioloba*, *Searsia pyroides* and *Celtis africana*.

Detailed descriptions of the soil texture and chemical composition around the measurement site are given in Table 2. The soil around the site is loamy sand. The soil organic carbon content was 0.93 % and the pH was 5.69.

### 2.2 Vegetation sampling

The vegetation surrounding the measurement station was classified based on land-use and vegetation structure (Figure 2), for which a detailed sampling was performed four times during the period from April 2011 to January 2012. Seven homogeneous land-use units were identified, including maize fields, dry sandy grassland, moist sandy grassland, disturbed grassland, thornveld savannah, woodland savannah and plantation. Grasslands had a tree cover of less than 15%, whereas in savannas it was greater than 15%. Furthermore, savannas were divided into thornveld, which had sparsely distributed large trees, and woodland with a tree cover greater than 50%.

The central region of each homogeneous unit was sampled along a transect, which resulted in 42 plots in total, six per transect. Each of these plots had an area of 100 m². All of the plant species were counted and identified up to species level,





and their major growth form was recorded. In addition, for all the woody species higher than 1m, the mean height, mean canopy height and width of each species were determined.

Within each plot, leaf material was collected from four 1 $m^2$ subplots during each of the four periods, and a one-sided leaf area index (LAI) of all the leaves was determined for grasses, forbs, shrubs and trees. After this all the plant material was dried for three days and weighed.

## 2.3 Instrumentation

At the measurement site there are continuous measurements of atmospheric aerosols, trace gases and meteorology (Booyens

et al., 2015; Jaars et al., 2014; Laakso et al., 2013; Vakkari et al., 2014, 2015). In this paper, we present the data directly relevant to ecosystem dynamics from September 2010 to August 2013.

The meteorological measurements included air temperature and pressure, wind speed and direction, relative humidity and temperature gradient between 2 m and 8 m heights. Precipitation was measured by two tipping buckets (Vaisala and Casella) working in parallel. Radiation measurements were placed at a 3 m height and included incoming and outgoing PAR by Kipp

& Zonen PAR-lite sensors, direct and reflected global radiation by Kipp & Zonen CMP-3 pyranometers and net radiation by Kipp & Zonen NR-lite2 net radiometer.

Soil moisture and temperature were measured in one soil profile, having sensors at depths of 5, 20 and 50 cm. There was also a separate sensor approximately 10 m away from the soil profile. Soil temperatures were measured with PT100 platinum thermometers, soil moisture with Delta-T sensors, and soil surface energy flux with Hukseflux HFP01 heat flux plate at a 5

cm depth.

Carbon dioxide, water vapour, sensible heat and momentum fluxes were measured using an eddy covariance setup similar to the one described by Aurela et al. (2009). The sonic anemometer was a METEK USA-1, and the $CO_2$ concentrations were measured using a Li-Cor LI-7000 gas analyzer. The sampling frequency of these instruments was 10 Hz. The anemometer and the gas sampling tube were installed at a 9 m height, which was well above the average tree height of 2.5 m. The flow

rate of the sampling system was 6 L min$^{-1}$.The length of the inlet tube for the LI-7000 gas analyzer was about 20 m. The gas analyzer was calibrated every month with a high-accuracy $CO_2$ span gas, and synthetic air with a zero $CO_2$ concentration was continuously used as a reference gas.

The state of the measurement system was continuously monitored by visiting the measurement site once to twice a week. During each visit, the state of the measurements was logged and corrective actions taken as required. During the data

analysis the log file was used to check erroneous measurement periods.

## 2.4 Processing of eddy covariance data

The turbulent fluxes were calculated as 30 min block averages from the 10 Hz raw data after a double rotation of the wind coordinates (McMillen, 1988) and calculation of the $CO_2$ mixing ratios with respect to dry air by accounting for water vapour fluctuations (Webb et al., 1980). For each averaging period, the time lag between the anemometer and gas analyzer



signals was determined using a maximum covariance method. The fluxes were corrected for systematic losses using the transfer function method of Moore (1986). This included a compensation for the low-frequency losses due to block averaging. For the high-frequency losses, an empirical first-order transfer function representing the overall system performance was determined from the field data using the sensible heat flux as a reference. A spectral half-power frequency of 1.6 Hz was determined for $CO_2$. Idealized cospectral distributions (Kaimal and Finnigan, 1994) were assumed, providing

correction factors as a function of wind speed and atmospheric stability. The storage flux was calculated by assuming a uniform distribution of $CO_2$ between the soil surface and the measurement height. During the last measurement year, the $CO_2$ concentration was also measured at a 1.5 m height, which enabled us to calculate a two-point estimate for the storage flux. The calculated fluxes of $CO_2$ were filtered by discarding the data with a friction velocity lower than 0.2 m s$^{-1}$. There was only one longer period of malfunction of the gas analyzer, which lasted for 15 days in November 2010. In total, 33% of the $CO_2$

flux values in the final time series were missing or discarded, which is similar to a mean value of 35% of missing data (19-site average)  (Falge et al., 2001).

The flux "footprint area" was estimated using an analytical function introduced by Kljun et al. (2004). It was estimated that 90% of the flux originated within a distance of 324 m upwind from the measurement tower. Figure 2 shows circles with the radius of the mean 50% and 80% cumulative flux footprint. The real flux footprint is not a circle and it varies by being larger

during the night and smaller during the daytime. The footprint area is a homogeneous thornveld (Figure 2).

**2.5 Partitioning of the net $CO_2$ flux**

There are several approaches to partitioning the measured net $CO_2$ flux to its GPP and respiration components. In this study, the partitioning of the flux was tested with both night-time and daytime based methods. Night-time and daytime periods

were separated by a threshold PAR value of 20 µmol m$^{-2}$ s$^{-1}$.  In the "night-time method" only night-time data is used to directly determine a respiration function, and GPP is calculated as a difference between NEE and respiration. Three different respiration functions were tested: an exponential temperature function (Lloyd and Taylor, 1994), an exponential temperature function with a soil moisture effect (Reichstein et al., 2003) and a general Poisson function (Archibald et al., 2009). The Poisson function describes both the exponential increase of respiration with the temperature and its decrease at higher

temperatures.

The "daytime method" was tested by fitting a model to the daytime NEE measurements. The model parameters are the initial slope of light response, the $CO_2$ uptake at light saturation, vapour pressure deficit (VPD) and respiration. The details of this approach are described in Tagesson et al. (2015).

For both the daytime and night-time methods, the fit parameters were calculated in a moving window which was defined for

every day. The data set did not have gaps longer than 15 days, and thus the moving window was expanded from 6 to 20 days in order to cover at least 50 measurement points. The fitting of all model functions was done using "lsqnonlin" command in MATLAB Release 2015b, which uses a trusted region least square algorithm.



We did the partitioning using algorithm by Lloyd & Taylor (1994). In this algorithm the night-time respiration was fitted using the exponential temperature function:

$$R_{eco} = R_b \exp\left( E_0\left(\frac{1}{T_0} - \frac{1}{T_{soil} - T_1}\right)\right)$$ (1)


where $R_b$ is the base respiration, $E_0$ is the temperature sensitivity, $T_0$ = 56.02 K, $T_{soil}$ is the soil temperature at 5 cm depth and $T_1$ = 227.13 K (Lloyd and Taylor, 1994). This function was fitted in two steps by first fitting the $E_0$ parameter individually for each year and then fitting the $R_b$ parameter to each window separately.

The GPP values derived from the NEE observations and respiration estimates were used to fit the hyperbolic tangent

function of PAR for every day using the 6 to 20 days moving window:

$$F_p = F_{p,max} \cdot \tanh\left(\frac{d \cdot PAR}{F_{p,max}}\right)$$ (2)

where $F_{p,max}$ is the canopy assimilation at light saturation and $d$ is the initial slope of light response (von Stamm, 1994). The missing values in the GPP time series were filled with the GPP which was calculated using this function. Finally, the NEE measurements were gap filled using the sum of respiration and GPP.

The eddy covariance measurement data used in this study covered the period from September 2010 to August 2013. We analysed the data by growing seasons rather than by calendar years, as a growing season at this southern hemispheric site spreads to two consecutive years.

**2.6 Satellite data**

Monthly values of MODIS NDVI (MCD43A4, 500 m pixel) product were extracted for the flux footprint of the eddy

covariance measurement (Figure 2). This product uses the reflectance data which is adjusted using a bidirectional reflectance distribution function for view angle effects. The NDVI signal is a simple transformation of spectral bands without any bias from ground-based parameters (Huete et al., 2002). Therefore, it is better suited for studies of long-term variation in vegetation structure than the MODIS LAI or MODIS GPP products.

**3 Results and discussion**

Figure 3 shows a time series of incoming PAR, air temperature, precipitation, evapotranspiration and $CO_2$ flux for the full measurement period. The incoming PAR and temperature were highest in January and lowest in July. Precipitation was highest during the growing season 2010-2011. Heavy rain events were followed by high evapotranspiration events, and the total evapotranspiration was higher during the rainy season due to higher precipitation and transpiration rates. The peak





evapotranspiration events in different years were similar in magnitude but their frequency varied. The highest inter-annual

variation in $CO_2$ flux occurred during the wet season, whereas during the dry season $CO_2$ fluxes were rather similar in

magnitude.

Based on the vegetation sampling described above, the LAI within the flux footprint had a maximum value of 2.32 $m^2 m^{-2}$ in

April and a minimum of 0.37 $m^2 m^{-2}$ in July (Table 3). The leaf biomass followed the same trend, with a maximum value of

644 $g m^{-2}$ and a minimum of 233 $g m^{-2}$ (Table 4).

## 190 3.1 Partitioning of the net $CO_2$ flux

The daytime canopy carbon assimilation in the middle of the wet season (DJF) followed the common pattern where the

daytime $CO_2$ flux increased linearly with increasing incoming PAR until it reached a saturated value (Figure 4). The dry

season (JJA) $CO_2$ flux rates were an order of magnitude lower than the wet season rates and canopy assimilation was rarely

saturated with respect to PAR. The "daytime method" fitting (Tagesson et al., 2015) was not successful because all

parameters had unrealistically high values, and thus the modelled respiration values were unrealistic.

The night-time respiration did not show a clear exponential increase with either soil moisture or soil temperature in any of

the respiration fitting windows, which ranged from 6 to 20 days (data not shown). The highest ranges of soil moisture in

individual fitting data sets were from 1 to 7% during the dry season and from 8 to 21% during the wet season, but there was

no significant relationship with respiration. A clear linear increase in respiration with increasing soil moisture was only

observed once, after the first intense rainfall event in early November 2010. Similarly, the night-time respiration did not

increase strongly with soil temperature in any of the respiration fitting windows. Instead, the respiration rate remained rather

constant with a ~2 $\mu mol m^{-2} s^{-1}$ reduction at the highest temperatures during the middle part of the wet season in 2010-2011.

However, on an annual time scale, higher respiration rates were observed when soil temperatures were higher (Figure 5).

There is little correlation with soil water content during either dry or wet season, or on an annual time scale. Therefore, it

seems that the ecosystem respiration is driven by plant phenology, being higher during the rainy seasons and on average

unaffected by short-term variations in soil water content and temperature. Our respiration relations are similar to those

reported by Tagesson et al. (2015), who observed no relationship between the night-time NEE and the environmental drivers

for 7-day periods in grazed savanna in Senegal.

The exponential temperature function with soil moisture effect (Reichstein et al., 2003) was successfully fitted to each

window, but a large part of the respiration values calculated with the resulting parameters were negative. In addition, the

correlation between the modelled and measured respiration rates was poor ($R^2=0.11$). The fit based on the Poisson function

(Archibald et al., 2009) was not successful either. Due to these findings the night-time respiration for gap filling was

determined using the exponential temperature function by Lloyd and Taylor (1994), Eq. (1). Even though the temperature

dependency was weak, the respiration rates modelled with this function correlated well with the measured respiration

($R^2=0.56$).



The annual NEE for the growing season 2012-2013 using the two-point estimate for the storage flux was 88 g C m$^{-2}$, which was 51 g C m$^{-2}$ less than that balance obtained with one-point storage estimates. The one-point storage flux was used for the whole measurement period because the two-point concentration data covered only the last year of measurements. Assuming that the difference between the storage calculation methods leads to similar differences in the annual NEE, the NEE of the 220 growing season 2011-2012 remains slightly positive, while that of 2010-2011 becomes even more negative.

### 3.2 Diurnal CO$_2$ cycle

The mean monthly diurnal variation of NEE reveals a change in the ecosystem dynamics during the transition from the dry to wet season (Figure 6, Figure 7). The highest values of carbon uptake occurred every year in December or January at 10 225 am to 12 am, whereas the incoming PAR had its peak between 11 am to 1 pm. This phase difference is caused by the stomata closure before the radiation peak in order to avoid water losses.

In September the mean diurnal NEE cycle can be depicted by a smooth curve with a nearly levelled maximal uptake period from 10 am to 3 pm. In November the diurnal pattern had a sharp dip and a levelling after the maximal uptake was reached. The peak NEE value in November 2012 was reached already at 9 am whereas in the other years the November peak NEE 230 value was reached at 11 am. Later on, during the middle part of the wet season, the afternoon dip was reduced and the diurnal pattern was closer to that observed in September.

The daytime NEE values were found to decrease with increasing VPD above a 2 kPa limit. This response was also found in all C$_3$-plant-dominated sites in a previous 11-site inter-comparison between African ecosystems (Merbold et al., 2009).

### 3.3 Seasonal variation of the carbon balance

In this section, we analyze the differences in carbon fluxes and their environmental drivers at a monthly scale during the three measurement years. Figure 8 shows the monthly sums of respiration, GPP, NEE, and precipitation together with the monthly averages of air temperature, soil temperature, soil moisture and daily maximum VPD.

In the growing season 2010-2011 the carbon uptake was the highest. Heavy rainfall occurred in November to January and soil temperature was relatively low. This contributed to the strong growth of vegetation and net carbon uptake. Furthermore, 240 the low values of VPD from January to April resulted in continued uptake of carbon. Due to the relatively low values of soil temperature throughout the growing season, the yearly respiration is lower than during the year 2012-2013.

During the year 2011-2012 the annual rainfall was the lowest in comparison with all the other years. Specifically, the December-January precipitation was relatively low. Moreover, the VPD was relatively high during this period and thus the growth of vegetation was low. The total LAI was 1.32 in January 2012 (Table 3). In February and March, the carbon balance 245 was positive possibly due to the growth of heterotrophic respiration and relatively small LAI.

During the year 2012-2013 the respiration was relatively high already in September. The early rainfall may have lead to the early growth of soil microbial population that enhanced the soil respiration capacity for the whole season. Furthermore, the





respiration was high throughout the growing season due to consistently higher air temperature and soil temperature. From December to February the monthly GPP was relatively high but the net carbon uptake was less than during the year 2010-2011 due to the much higher respiration. From February to April the soil moisture was relatively low and VPD was high and thus the carbon uptake by photosynthesis was limited. This led to the most positive carbon balance of all years.

There was roughly a ten-fold difference in the monthly GPP sums between the wet and dry seasons. During the dry season, the grasses are dormant and only trees are contributing to GPP. During the dry season, this contribution did not vary between the years, even though soil moisture varied significantly. Therefore, the inter-annual variation in GPP is largely due to the variation during the wet season.

At a monthly time scale there were two periods which largely determined the inter-annual variation of NEE. Firstly, at the beginning of the rainy season (September to November) NEE showed a large variation which was due to variation in both the respiration and GPP components. Secondly, from December to January the ecosystem was taking up carbon during each year. During this period the monthly respiration and GPP were highest and the monthly respiration followed precipitation. The primary carbon uptake period spanned from December to January whereas the total carbon uptake period varied in magnitude and in length from 3 months to 6 months.

In conclusion, the high precipitation in December and January led to large carbon uptake during the growing season. On the other hand, early rain in September and relatively high air and soil temperatures throughout the year resulted in significantly higher yearly respiration and thus more positive carbon balance.

**3.4 Inter-annual variation of the carbon balance**

The carbon balances for the years 2010-2011, 2011-2012 and 2012-2013 were -85, 67 and 139 g C m$^{-2}$ y$^{-1}$, respectively (Table 5). The changes in the yearly NEE sum are not explained by the changes in annual precipitation, temperature, length of rainy season or peak NDVI. However, the number of days when soil moisture was higher than 7% is related to the annual NEE sum. The soil moisture limit of 7% is thought to be critical limit under which plants are water stressed (Archibald et al., 2009). Given that the data covered only three years, it might be difficult to generalize the relation between the annual carbon balance and the number of wet soil days. Moreover, at monthly and weekly scale the carbon balance and wet soil days are not correlated.

The annual GPP variation followed the variation in precipitation and peak NDVI. The precipitation response was also found in a previous 11-site inter-comparison between African ecosystems (Merbold et al., 2009).

These results show that it might not be possible to relate the annual carbon balance to any set of annual aggregates of the environmental drivers. As shown in the previous section, the environmental drivers like soil moisture and soil temperature do partly control the wet season carbon fluxes whereas during the dry season the carbon balance is less sensitive to the changes in these variables. Therefore, the environmental drivers can have different kind of effect on carbon balance during different seasons. Furthermore, the annual sum of NEE is a small difference of two large components of carbon uptake by





photosynthesis and the release of carbon to the atmosphere by respiration and thus the NEE sum is sensitive to small changes
      in its components.

**3.5 NDVI as proxy for GPP**

There was a strong positive correlation between the monthly sum of GPP and the monthly mean of NDVI ($R^2$=0.83,
p<0.001) (Figure 9). Sjöström et al. (2009) found also a high correlation between 8-day NDVI and GPP sum in Sudanian
savanna with a canopy cover comparable to the Welgegund site. The deviations of the regression and observation based
      GPPs were largest during the middle of the wet season, when NDVI underestimated GPP, and during the early dry season,
      when NDVI overestimated GPP.

Figure 10 shows a 12 year time series of monthly NDVI and precipitation data. The peak value and the shape of the annual
cycle of both variables vary significantly between the years. The peak NDVI occurred each year between December and
March and it lagged the peak rainfall by one or two months with the exception of the year 2007. Within our flux data period,
      the peak NDVI was similar in 2011-2012 and 2012-2013, because it did not capture the rainy season peak in GPP in 2012-
      2013. On the other hand, it did highlight the year 2010-2011, which had a significantly higher GPP. Based on the NDVI data,
      the year 2010-2011 represents a common pattern at this site, whereas the years 2011-2012 and 2012-2013 have lower NDVI
      peak values.

Scanlon et al. (2002) demonstrated with a 16 year NDVI time series from the Advanced Very High Resolution Radiometer
      across a rainfall gradient that grassy areas contributed most to the inter-annual variation in NDVI. In addition, trees have
      been shown to have more consistent phenological cycles in savannas (Archibald and Scholes, 2007). Therefore, according to
      NDVI, the Welgegund site shows characteristics of a grassland.

From September 2001 to August 2013, the yearly maximum values of the measurement site NDVI were 0.02 units smaller
on average than a nearby moist sandy grassland area which is not grazed (land-use class 6 in Figure 2). This difference is
      most probably due to heavy grazing at the measurement site.

**3.6 Comparison to other sites**

The annual NEE sum and its inter-annual variation at our site differs significantly from the results reported in a previous
study at a grazed savanna grassland in Dahra, Senegal (Tagesson et al., 2015). The Dahra site had a peak MODIS LAI
      (MOD15A2) between 1.4 to 2.1 and a mean annual precipitation of 524 mm, which are similar to the Welgegund site. At
      Dahra, the yearly carbon balance varied from -336 to -227 gC m$^{-2}$ year$^{-1}$ during three years. The major difference between
      these two sites is that the dominant grass species change yearly at Dahra, whereas Welgegund has a perennial grass layer.
      The high grazing pressure at Welgegund and large amount of missing data at Dahra may explain the large difference in the
carbon balance between these sites.





On the other hand, the carbon balance at our site is similar to the balance measured at Skukuza site in Kruger National Park, South Africa, which has similar precipitation to Welgegund but a significantly higher tree cover of 30% (Archibald et al., 2009). The reason for this agreement in carbon balance is probably due to large mammalian herbivore population and fires which contribute to the more positive carbon balance at the Skukuza site.

At the savanna sites of Nolohou, Benin and Bontioli, Burkina Faso which have significantly higher annual precipitation (852 to 1190 mm/yr) and higher LAI values, the carbon balance varied from -429 to -136 g C m$^{-2}$ yr$^{-1}$ (Ago et al., 2014; Brümmer et al., 2008). These sites were not grazed but the grasses were burned annually. During the dry season at the Nolohou site, higher soil moisture resulted in higher soil respiration and thus more positive total carbon balance. In contrast, at the Bontioli site higher precipitation during the transition period from wet to dry season resulted in higher uptake of carbon. However, at

Welgegund the dry season fluxes are order of magnitude smaller than the wet season fluxes and the dry season carbon balance did not vary between the years. Similarly, the dry season carbon balance did not have significant differences at grassland, crop land and nature reserve in the Sudanian Savanna which receives similar amount of precipitation as the Welgegund site (Quansah et al., 2015).

**4 Conclusion**

The results of this study indicate that the inter-annual variation of NEE is high at the Welgegund savanna grassland site compared with a grazed savanna grassland in Senegal. The carbon balances for the years 2010-2011, 2011-2012 and 2012-2013 were -85, 67 and 139 g C m$^{-2}$ y$^{-1}$, respectively. This compares with the variation at the Kruger National Park where the annual NEE ranged from -138 to 155 g C m$^{-2}$ y$^{-1}$ during a 5-year measurement period (Archibald et al., 2009).

The night-time respiration was not significantly dependent on either soil moisture or soil temperature on a weekly temporal

scale, whereas on an annual time scale higher respiration rates were observed when soil temperatures were higher. The lack of functional dependence on soil moisture and temperature differs from the findings of previous studies and highlights the importance of testing various partitioning methods for the flux data from savannas.

The beginning of the rainy season (September to November) and the mid growing season (December to January) largely determined the inter-annual variation of NEE. Early rainfall in September 2012 and higher soil temperature resulted in

higher respiration and thus more positive carbon balance. During the mid-growing season the respiration and GPP were highest and the monthly respiration followed precipitation.

Future work should focus on the respiration variations during the daytime. With more direct soil respiration measurements and cattle respiratory flux measurements the overall uncertainty of the total ecosystem respiration estimates could be reduced. In addition, these measurements would provide much needed information about the environmental drivers of

respiration specific to savanna ecosystems.



**Acknowledgements**

This work was supported by Finnish Meteorological institute, The North-West University and University of Helsinki, and the Finnish Academy project *Developing the atmospheric measurement capacity in Southern Africa* and Finnish Centre of Excellence, Grant no. 272041. The authors wish to thank Eduardo Maeda for downloading and processing the MODIS
NDVI data, South African Weather Service (SAWS) for the provision of the long-term rainfall and temperature data and the farmers at the ranch.

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

**Tables**

**Table 1: Dominant species at the flux footprint.**

| Species | Growth form | Mean leaf area (cm²) | Mean height (m) | Mean canopy (m) | Number of individuals |
|---|---|---|---|---|---|
| *Vachellia erioloba* (E.Mey.) P.J.H.Hurter | Tree | 5.9 | 3.2 | 1.8 | 16 |
| *Searsia pyroides* (Burch.) Moffett | Tree | 8.2 | 2.9 | 1.2 | 9 |
| *Celtis africana* Burm.f. | Tree | 18 | 3.4 | 1.4 | 6 |
| *Ehretia rigida* (Thunb.) Druce | Tree | 4.1 | 2.5 | 1.8 | 5 |
| *Vachellia karroo* (Hayne) Banfi & Galasso | Tree | 9.4 | 2.4 | 1.6 | 3 |
| *Diospyros lycioides* Desf. | Shrub | 7.9 | 1.8 | 0.8 | 7 |
| *Asparagus laricinus* Burch. | Shrub | 0.7 | 1.5 | 1.1 | 6 |
| *Asparagus suaveolens* Burch. | Shrub | 0.5 | 1.3 | 0.9 | 4 |
| *Grewia flava* DC. | Shrub | 6.7 | 2.2 | 1.8 | 3 |
| *Pentzia globosa* Less. | Forb | | < 0.5 | | 18 |
| *Walafrida densiflora* Rolfe | Forb | | < 0.5 | | 11 |
| *Hermannia depressa* N.E. Br. | Forb | | < 0.5 | | 8 |
| *Dicoma tomentosa* Cass. | Forb | | < 0.5 | | 6 |
| *Euphorbia inaequilatera* Sond. | Forb | | < 0.5 | | 4 |
| *Panicum maximum* Jacq. | Graminoid | | < 1.5 | | 18 |
| *Setaria sphacelata* (Schumach.) Stapf & C.E.Hubb. ex Moss | Graminoid | | < 1.5 | | 14 |
| *Eragrostis trichophora* Coss. & Durieu | Graminoid | | < 1.5 | | 11 |
| *Themeda triandra* Forssk. | Graminoid | | < 1.5 | | 10 |
| *Eragrostis curvula* (Schrad.) Nees | Graminoid | | < 1.5 | | 9 |




**Table 2: Soil structural and chemical composition of each homogeneous land-use unit. MF, Maize fields; WL, Woodland; TV, Thornveld; P, Plantation; DSG, Dry sandy grassland, MSG, Moist sandy grassland, DG, Disturbed grassland.**

|  | MF | WL | TV & P | DSG | MSG | DG |
|---|---|---|---|---|---|---|
| Medium-Coarse sand | 43.1 | 49.1 | 45.1 | 47.2 | 37.9 | 44.1 |
| Fine sand | 44.0 | 37.8 | 40.1 | 37.7 | 41.3 | 45.6 |
| Silt | 3.5 | 7.6 | 7.8 | 8.0 | 8.5 | 3.4 |
| Clay | 9.4 | 4.7 | 7.0 | 7.2 | 12.3 | 7.0 |
| Org. cont. (%) | 3-6 | 3-6 | 3-6 | 3-6 | 3-6 | 3-6 |
| Organic-C (%) | 0.68 | 0.87 | 0.93 | 0.91 | 1.01 | 0.54 |
| Acidity (pH) | 5.12 | 5.73 | 5.69 | 5.09 | 6.52 | 5.22 |
| CEC (cmol(+)/kg) | 9.56 | 11.63 | 12.1 | 11.37 | 12.42 | 10.42 |
| S-value (cmol(+)/kg) | 2.63 | 4 | 3.42 | 2.29 | 9.51 | 1.05 |
| Base saturation (%) | 27.52 | 34.38 | 28.27 | 20.12 | 76.62 | 10.05 |
| Nitrogen (%) | 0.01 | 0.06 | 0.07 | 0.07 | 0.08 | 0.03 |
| Calcium (mg/l) | 373 | 605 | 449 | 267 | 1009 | 108 |
| Magnesium (mg/ l) | 74 | 97 | 101 | 69 | 477 | 37 |
| Potassium (mg/l) | 337 | 230 | 349 | 365 | 583 | 253 |
| Phosphorus (mg/l) | 3 | 3.54 | 3.31 | 2.89 | 3.42 | 2.78 |
| Sodium (mg/l) | 0.5 | 7.5 | 4.5 | 6.5 | 26 | 5.5 |
| Sulphur (mg/l) | 16.4 | 13.88 | 14.39 | 12.95 | 13.16 | 11.54 |
| Arsenic (mg/kg) | 1.074 | 1.06 | 1.213 | 1.436 | 1.402 | 1.2 |
| Aluminium (mg/kg) | 8205 | 4655 | 5993 | 6705 | 9517 | 5963 |
| Cadmium (mg/kg) | 0.016 | 0.026 | 0.022 | 0.026 | 0.028 | 0.017 |
| Chromium(mg/kg) | 163 | 48 | 110 | 84 | 111 | 86 |
| Cobalt (mg/kg) | 10.26 | 5.32 | 8.97 | 4.29 | 10.1 | 3.35 |
| Copper (mg/kg) | 11.83 | 7.97 | 9.85 | 9.3 | 13.03 | 7.16 |
| Iron (mg/kg) | 16365 | 9965 | 14778 | 12448 | 17313 | 11168 |
| Lead (mg/kg) | 3.28 | 3.34 | 3.51 | 3.86 | 5.3 | 3.1 |
| Manganese (mg/kg) | 465 | 350 | 395 | 218 | 625 | 157 |
| Nickel (mg/kg) | 38 | 11 | 29 | 18 | 30 | 10 |
| Zinc (mg/kg) | 9.9 | 8.32 | 7.36 | 9.57 | 11.15 | 6.21 |

**Table 3: One-sided leaf area index within the flux footprint area in 2011-2012.**

|  | Year | Months | Sampling month | Herbaceous LAI | Woody LAI | Total LAI [$m^2 m^{-2}$] | Herbaceous LAI/ Total LAI [%] |
|---|---|---|---|---|---|---|---|
| **Autumn** | 2011 | March-May | April | 1.20 | 1.12 | 2.32 | 51.7 |
| **Winter** | 2011 | June-August | July | 0.12 | 0.25 | 0.37 | 32.4 |
| **Spring** | 2011 | September-November | October | 0.30 | 0.47 | 0.77 | 39.0 |
| **Summer** | 2012 | December-February | January | 0.53 | 0.79 | 1.32 | 40.2 |




**Table 4: Leaf biomass within the footprint flux area in 2011-2012.**

|  | Year | Months | Sampling month | Herbaceous [g m$^{-2}$] | Woody [g m$^{-2}$] | Total [g m$^{-2}$] |
|---|---|---|---|---|---|---|
| **Autumn** | 2011 | March-May | April | 383 | 261 | 644 |
| **Winter** | 2011 | June-August | July | 147 | 86 | 233 |
| **Spring** | 2011 | September-November | October | 108 | 169 | 277 |
| **Summer** | 2012 | December-February | January | 157 | 224 | 381 |


**Table 5: Annual carbon balance and key environmental drivers calculated for each year from September to the August of the following year.**

|  | NEE [gC m$^{-2}$ yr$^{-1}$] | GPP [gC m$^{-2}$ yr$^{-1}$] | Respiration [gC m$^{-2}$ yr$^{-1}$] | Annual precipitation [mm] | Rainy season length [days] | Peak NDVI | Number of days when soil moisture was higher than 7 % | Annual total PAR [mol m$^{-2}$] |
|---|---|---|---|---|---|---|---|---|
| 2010 | -85 | -1360 | 1275 | 721 | 207 | 0.53 | 232 | 15500 |
| 2011 | 67 | -1014 | 1080 | 422 | 178 | 0.43 | 101 | 15300 |
| 2012 | 139 | -1179 | 1318 | 615 | 228 | 0.44 | 79 | 15500 |

**Figures**







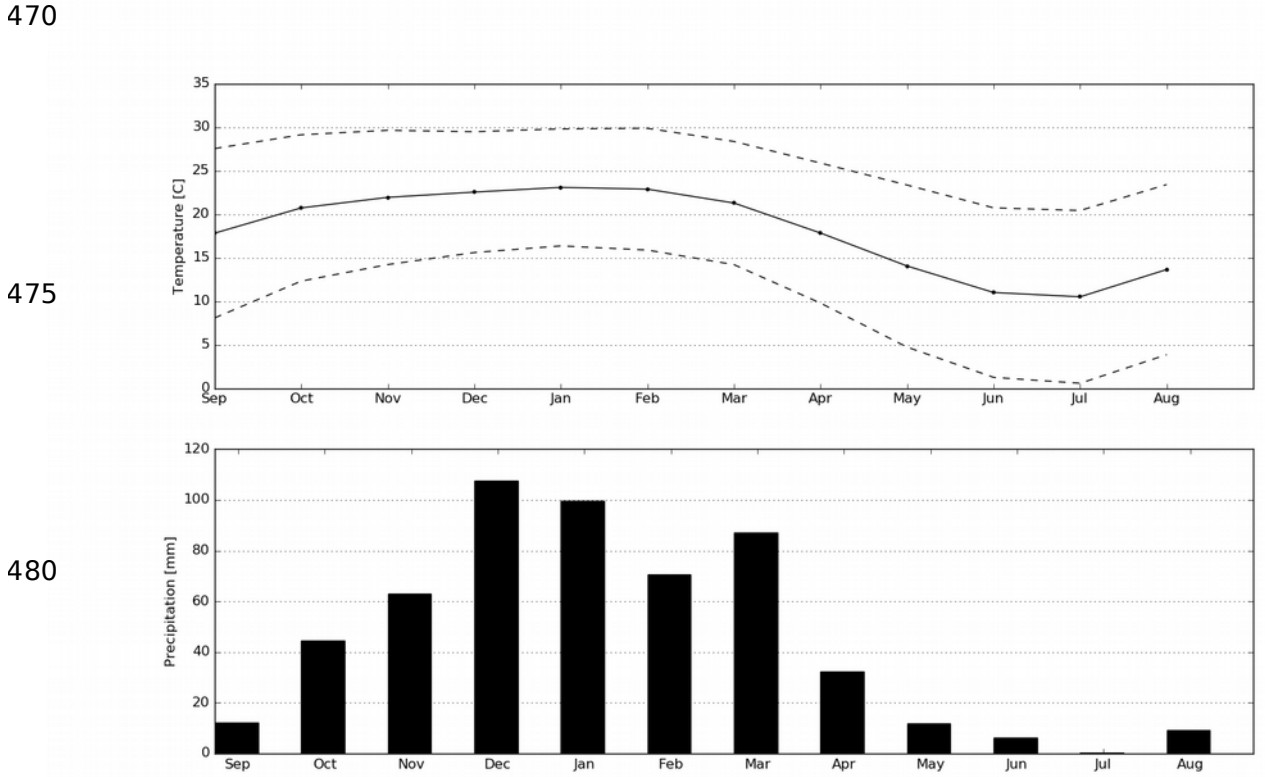

**Figure 1: Monthly mean meteorological data from a nearby weather station in Potchefstroom during 1998-2014. The upper figure shows the mean (solid line) and minimum and maximum (dashed lines) temperatures. The lower figure shows the mean precipitation.**






**Figure 2: Satellite image of the Welgegund measurement site showing the seven different sampling regions based on different vegetation and land-use classes. The red square shows a MODIS NDVI pixel at 500 m spatial resolution. The blue circles indicate the diameter of the mean 50% and 80% cumulative flux footprint.**






**Figure 3: Time series of incoming PAR, air temperature, precipitation, evapotranspiration and CO$_2$ flux for the measurement period from September 2010 to August 2013. The solid lines within the CO$_2$ flux data show the five day centered mean of minimum daytime and maximum night-time CO$_2$ flux. Incoming PAR, air temperature and CO$_2$ flux data are 30 minute averages.**





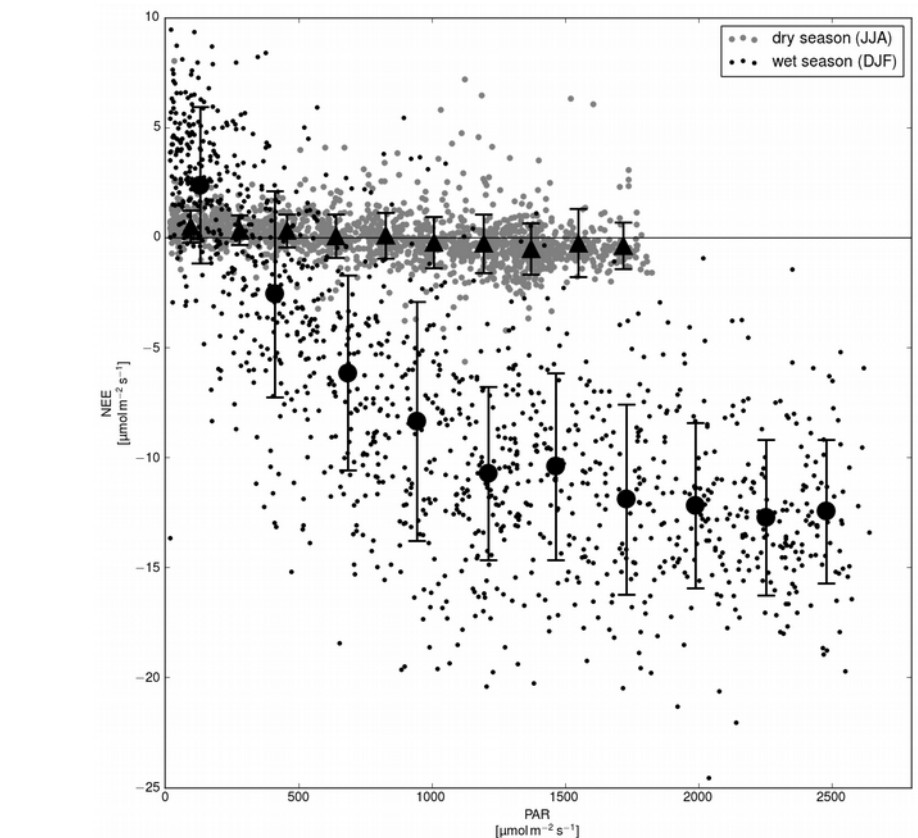

**Figure 4: Relationship between PAR and daytime NEE for wet (DJF) and dry (JJA) seasons from September 2010 to August 2011. The triangles (dry season) and circles (wet season) indicate bin averaged values and error bars indicate ±1 standard deviation. Daytime was defined as periods when PAR is higher than 20 μmol m⁻² s⁻¹.**





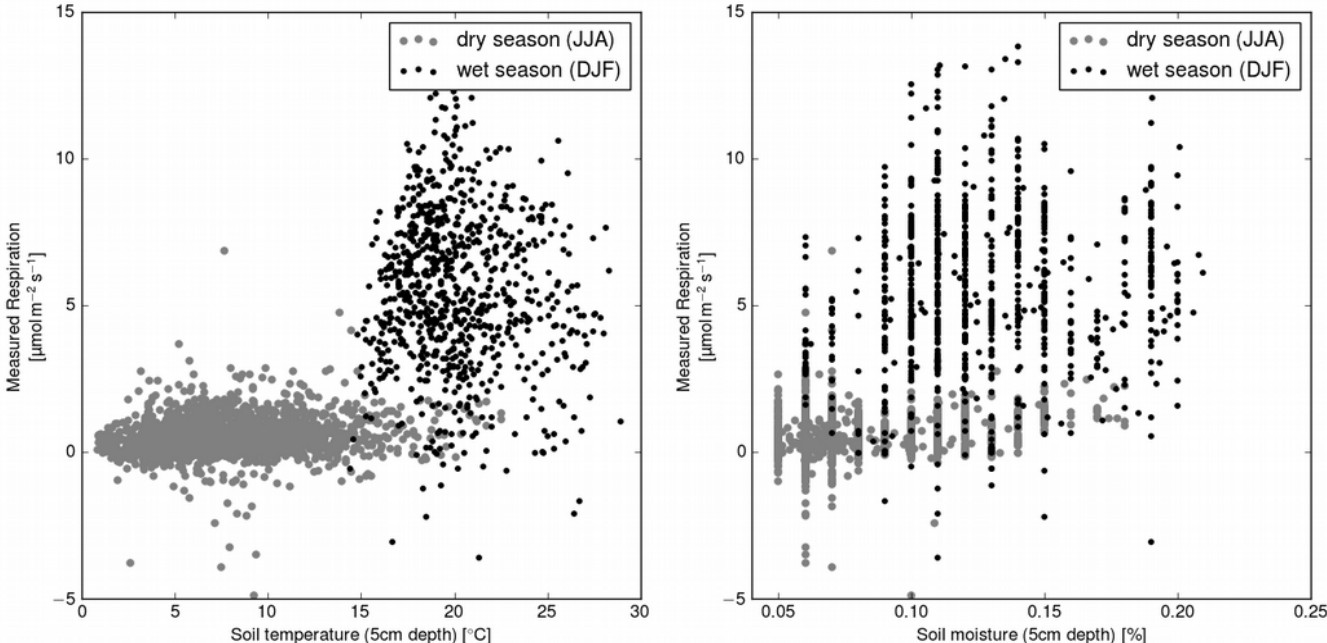

**Figure 5: Relationship between night-time respiration and soil temperature (left) and soil moisture (right) for wet (DJF) and dry (JJA) seasons from September 2010 to August 2011. Night-time was defined as periods when PAR is less than 20 μmol m$^{-2}$ s$^{-1}$.**




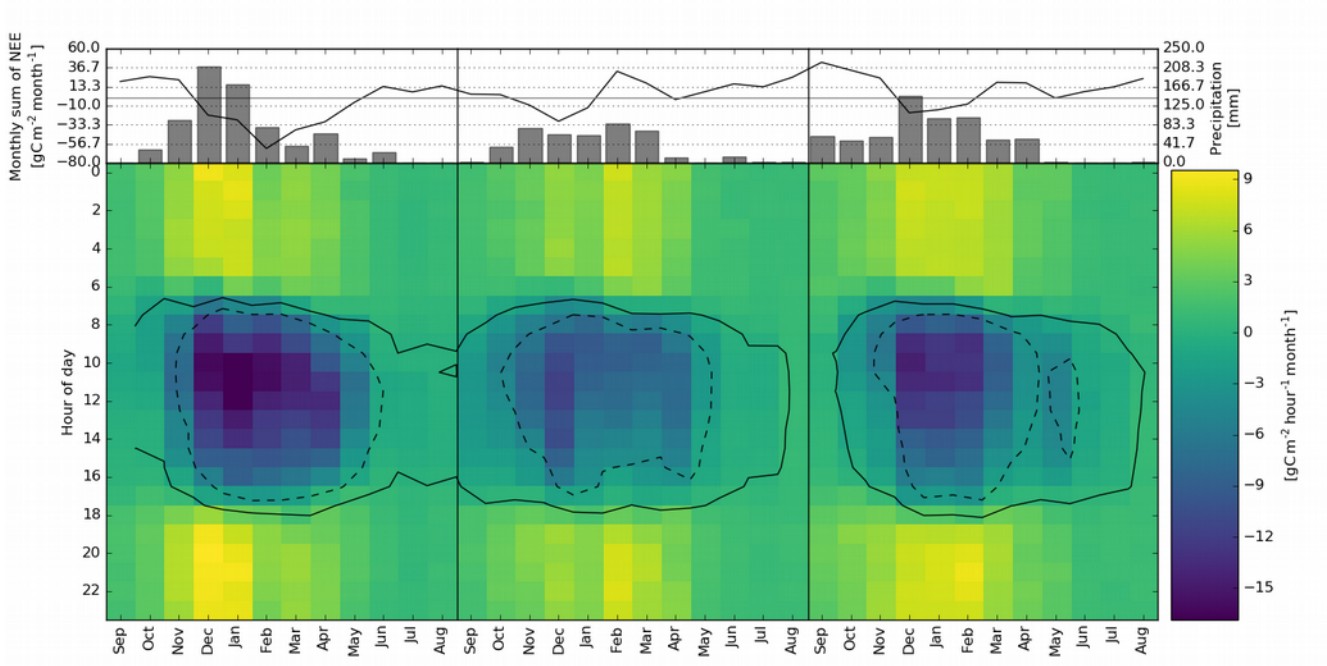

**Figure 6: Contour plot of monthly NEE sums for each hour of day. The solid line shows the zero isoline and the dashed line shows**
**the NEE values less than -4 g C m$^{-2}$ h$^{-1}$ mo$^{-1}$. The upper panel shows the monthly NEE as a solid line and the monthly precipitation as bars.**




**Figure 7: Monthly diurnal cycle of NEE for the early and mid-season months of the wet season in 2010, 2011 and 2012.**







**Figure 8: Monthly sums of respiration, GPP, NEE, precipitation and the monthly average of air temperature, soil temperature, soil moisture and daily maximum VPD. The panels on the right side show the yearly sums for the years 2010-2011, 2011-2012 and 2012-2013.**




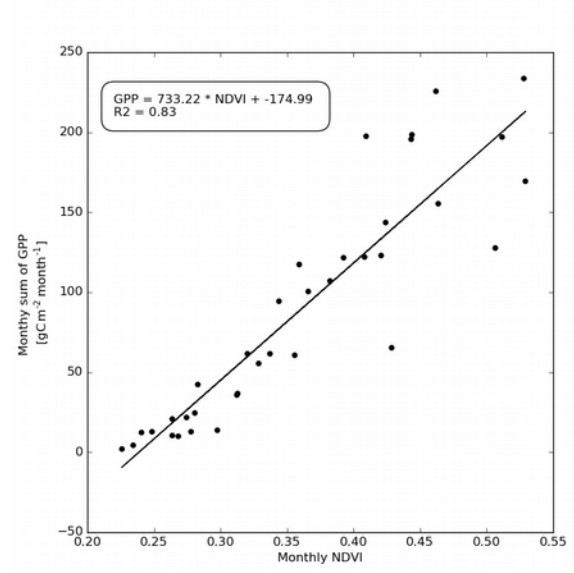



Figure 9: Linear regression between monthly NDVI and monthly sum of GPP.







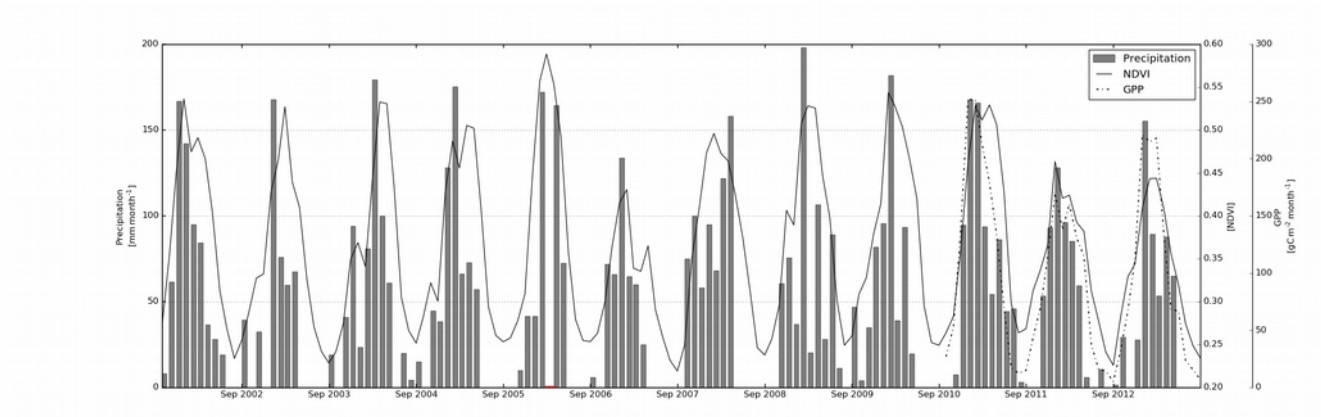

**Figure 10: Time series of monthly precipitation (bars), NDVI (solid line) from September 2001 to August 2013 and GPP from**
**September 2010 to August 2013. The precipitation was measured at a nearby weather station (SAWS) in Potchefstroom. Red axis**
**denotes missing value of precipitation in February 2006.**



