# Peer review of "Carbon balance of a grazed savanna grassland ecosystem in South Africa"

_Biogeosciences, 2016_

## Referee Comment (RC1) · Anonymous Referee #1 · 1 Aug 2016

The paper by Räsänen et al. explores carbon dioxide fluxes measured with the eddy covariance method for three years at a grazed savanna grassland in Welgegund, South Africa. The material is appropriate for a scientific study and the data obtained appear to be high-quality. It is relevant for many African ecosystems to focus on CO2 fluxes response to environmental drivers in order to better predict fluxes patterns in the context of climate change. Therefore, the work is interesting and worthy of publication in Biogeosciences Journal because of the lack of knowledge regarding the carbon cycle for Africa continent. However, I have a number of issues with the paper which lead me to suggest that it requires major revisions before it becomes acceptable for publication in BG.

General Comments:

[Figure]

1) Firstly, while the study site is located on a savanna grassland which is grazed by cattle and sheep, authors did not provide any information on the average stocking rate and the management of the site during the studied period. Is the site grazed intensively or not? What was the stocking rate? How the grassland was managed? What is the slope of the field? At the measurement height what is the fetch? Was the fetch adequate to characterize the carbon dioxide and water vapor fluxes of the vegetation type? These are important for understanding and interpreting the results.

2) It is also well known (see references below) that grazing affects a range of ecological and biogeochemical processes and properties, including plan community composition, soil physical properties, soil C and nitrogen content and the magnitude of C and carbon dioxide exchanges which in turn influence soil organic carbon storage. This study could have been more attractive if the impact of grazing on carbon dioxide exchange had been investigated. This probably would help to better assess for example the relation between the total ecosystem respiration and environmental drivers.

3) Authors used the Kaimal cospectra in the computation of the correction factors that are used to correct the high frequency losses (L129 – 130). However, recent studies (Mamadou et al., 2016) showed that Kaimal cospectra can be significantly different from sensible heat cospectra, and the high-frequency loss correction for $CO_2$ using these different cospectra resulted in the large difference in $CO_2$ flux calculations, i.e., using Kaimal cospectra can result in an overestimation of $CO_2$ fluxes even if the site could not be considered as difficult (i.e., fairly flat, homogeneous, low vegetation, sufficient measurement height). Especially, at their studied site, authors found that the choice of Kaimal rather than sensible heat cospectra reversed the annual carbon balance from being a net C sink to being a weak C source. Did the authors verify if their kaimal cospectra differ or not from sensible heat cospectra before chosen them as idealized cospectra?

4) Most of results presented in the section 3.4 are too much qualitative, superficial and descriptive and should be supplemented with additional statistical analyses in order to

provide more quantitative rigor.

5) The uncertainties associated to the annual carbon dioxide balance estimation are not evaluated. This remains a great lack for the study. The authors also clearly mentioned in their introduction that environmental drivers for the inter-annual variation in NEE are poorly understood. Unfortunately no progress regarding this point has been made within the present study.

Specific comments

L18-19: What about the dependence, at monthly scale, of the nighttime respiration on soil moisture or soil temperature? L24-25: by increasing autotrophic respiration? L32: The seasonal cycle of what? Please clarify. L32-33: The alternation of "wet and dry seasons" cannot in my view be generalized for the "whole Africa". In other regions of Africa, the dry and wet seasons are separated for example by two transitional seasons... L67, in site description section: Please, give values of the roughness length, zero-displacement height and site's slope. L102-103: Specify the sampling rate of the meteorological variables. L113: Specify the type of the gas analyzer. L115-118: What are the characteristics of the sampling tube (inner diameter etc.), the pump and the gas used for the zero and span? L127: Give an indication of the magnitude of low frequency correction factors. L129-130: Provide an illustration of kaimal and the sensible heat cospectra according atmospheric stability to attest that both cospectra match. L133: Replace the calculated fluxes by "the corrected fluxes". L133: What was the fraction of data excluded this way? L133-136: Do you only use u* filtering criteria to discard bad data? if Yes, explain why. L181: Complete "air" with temperature. L182: You never indicated how water vapor data have been treated. What is the cut-off frequency for H2O fluxes? How these data have been corrected for low and high frequency losses? Which criteria have been used for the filtering of bad data? L183: Explain how high evapotranspiration rate were due to higher precipitation and transpiration rate during the rainy season? What about soil evaporation? L206: air or soil temperature? L211-215: The low (high) values of the correlation coefficients cannot only be used to attest

the robustness of dependences. These must be accompanied with the p-values. L225-226: showed how? L232: I cannot get this conclusion... L223–L233: Why is there so much interpretation in the results? L301: most or must? L305-310: I am afraid that because of the difference of their climate, the Dahra site and cannot be easily compared to the Welgegund site. You should mention this in your discussion. L315, L317: Write Nalohou not Nolohou... L475: Figure 1 and also in the title: "air" or "soil" temperature? L500: Figure 3: Is it necessary to show evapotranspiration curve? L522: Figure 4: bin averaged for how many data?

References Jérôme, Elisabeth, Yves Beckers, Bernard Bodson, Bernard Heinesch, Christine Moureaux, and Marc Aubinet. 'Impact of Grazing on Carbon Dioxide Exchanges in an Intensively Managed Belgian Grassland'. Agriculture, Ecosystems Environment 194 (1 September 2014): 7–16. doi:10.1016/j.agee.2014.04.021. Mamadou, Ossénatou, Louis Gourlez de la Motte, Anne De Ligne, Bernard Heinesch, and Marc Aubinet. 'Sensitivity of the Annual Net Ecosystem Exchange to the Cospectral Model Used for High Frequency Loss Corrections at a Grazed Grassland Site'. Agricultural and Forest Meteorology. Accessed 1 August 2016. doi:10.1016/j.agrformet.2016.06.008. Peichl, Matthias, Owen Carton, and Gerard Kiely. 'Management and Climate Effects on Carbon Dioxide and Energy Exchanges in a Maritime Grassland'. Agriculture, Ecosystems Environment 158 (1 September 2012): 132–46. doi:10.1016/j.agee.2012.06.001. Piñeiro, Gervasio, José M. Paruelo, Martín Oesterheld, and Esteban G. Jobbágy. 'Pathways of Grazing Effects on Soil Organic Carbon and Nitrogen'. Rangeland Ecology Management 63, no. 1 (January 2010): 109–19. doi:10.2111/08-255.1.

---

## Referee Comment (RC2) · Anonymous Referee #2 · 15 Aug 2016

Review of Carbon balance of a grazed savanna grassland ecosystem in South Africa by Rasanen et al. 2016

**General comments:**

The number of EC sites sampling Carbon fluxes at African savannas is very limited, and this manuscript thereby fills a very important gap in our knowledge regarding terrestrial C cycling. The manuscript thereby fits well into the objectives of Biogeosciences. It has recently been shown that grazing has a tremendous effect on $CO_2$ fluxes in savanna ecosystems, and it is thereby very important to measure fluxes in these ecosystems. However, the manuscript is written in a quite unclear way. It can easily be constrained at many parts, whereas many more details are needed in others.

Major concerns:

Generally, the language needs to be improved and the manuscript must be much more concise. This goes for all sections.

A large section of the results is about testing different partitioning methods, but it is not included as an aim in the introduction. In case an aim of the study is to investigate different partitioning methods, please clarify this already in the introduction. Otherwise, I would recommend using the one that works best. I fully understand why the night-time methods are not working well, since it has generally been seen that at a diurnal time scale respiration is not strongly linked to temperature for savanna ecosystems. But I cannot understand why the daytime method is working so poorly. Are you sure that you fitted the equation correctly? After putting a lot of effort into partitioning of the data, the partitioned data is not even shown except in the format of monthly averages. Please, show the partitioned data as well.

Why did you use monthly averages in the investigation of seasonal variation? I cannot see any reason for not using daily sums or averages. By using monthly estimates you hide a lot of the variability and it is more difficult to see the relationship to the environmental variables.

There is no statistical testing of relationships of the fluxes to the environmental variables. It is just stated that high flux values can be explained by some variables. But you do not explain how you have tested for this.

**Specific comments:**

L19-20. There is a contradiction in this sentence: are the balances yearly or are they for the growing season, please rephrase.

L23-24 Please clarify in the abstract why: This study underlines the difficulty in establishing a functional relation between the total ecosystem respiration and the environmental drivers in savanna ecosystems.

L24-25 There must be something that explains the inter-annual variability, even though it might be that you have not seen any explanations in your data sets.

L32 reference for the 20% of global area please.

L35. Please rephrase, it sounds like the humans are grazing.

L42 reference please

L45 . instead of ,

L51 This is not correct: Tagesson, Ago and Quansah is all sites affected by either grazing or agriculture, there are also EC towers in Wankama Falls, Agofou, and Demokeya, which are all affected by grazing or agriculture; see (Tagesson et al., 2016a)

L55 please include (Tagesson et al., 2016a) that investigated annual budgets for 6 different sites across the Sahel.

L61 I do not understand why you use NDVI as a proxy for GPP. You have EC measurements, why not use them directly? Please clarify.

L70 Do you have any data on number of sheep and cattle?

L86 please give exact sampling dates.

L91 What do you mean, did you count all plants inside the 100m$^2$ plot or did you identify all species?

L101 Are not all these measurements relevant to Ecosystem dynamics? Please use a different word than ecosystem dynamics.

L103 At 2 and 8 m height or at several heights between 2 and 8 m? What was the height of the tipping buckets?

L102 What sensors were used for the meteorological measurements?

L108 What sensor? what did it measure?

L115 Why 20 m when height of the sensor was only 9m, should be possible to have a much shorter tube than this. What sort of tube did you use, inner diameter? No filter between the IRGA and the incoming air? What was the separation length between the inlet tube and the anemometer?

L137-140 and Figure 2. The footprint is never this uniform for different wind directions, if you have estimated the footprint for each 30 min period; it would be easy to show an average for the different wind directions.

L140 If the footprint is homogeneous thornveld, why do you report vegetation sampling for all other vegetation types? Looking at figure 2 with the footprint, is seems like the only vegetation cover which is affecting the EC measurements are the thornveld. If you want to present all the other data as well, I think you should you must incorporate a reason for this in the introduction, and a link to the EC data.

L145-151 Please give equations for the all partitioning models.

L163 Why in two steps? Why give E0 an annual value and not using the moving window?

L166 What is Fp? Why fitting this at all? Why not using the light response function that was used in the "daytime method"?

L170 This is probably a good choice, but why not always use 1 September to 31 August or something similar? Why only estimating the growing seasons?

L174 This is incorrect MCD43A4 is not an NDVI product, it is a BRDF product, please rephrase. Why using monthly averages, when data is available as an 8 day product? It is not clear if you extracted the values of one single pixel, or did you use an average of several pixels?

L176 What do you mean by that NDVI is better to use than the LAI and GPP product for vegetation structure? First, you are not studying vegetation structure, you are studying fluxes. Different products are good for different things. I agree that NDVI is a useable parameter, but it is not a real value like LAI and GPP. You cannot claim that it is better to use than these other parameters for studying vegetation dynamics. It is so far very unclear what you are going to use these data for. Please clarify. I would state that NDVI is a proxy for vegetation phenology.

L180 please give sum of rain

L192 In figure 4, it does not look like a linear increase until the saturation level. It is rather asymptotic

L197 How could the parameters get unrealistically high? In case data looks like in Figure 4, parameters should be fine. In case the relationship is very linear, the saturated GPP level gets unrealistically high, but this does not really matter for the partitioning as long as the equation is well fitted to the data. As the manuscript is written now, it seems like you want to test several different partitioning methods, if this is the case then you must show how all these different partitioning methods differ in their output. In the results section, you do not show the output of the different methods at all.

L209 I would not say that it was fitted successfully in case the R2 value is 0.11.

L210-215, in case you want to make a proper comparison of the different partitioning methods, you should give statistics for all methods. Please also show a figure with modelled versus measured values. In order to make a proper comparison you must separate a part of the data set to be used for the model parameterisation and one part for the model evaluation. A suggestion would be to use a bootstrapping simulation methodology.

L216 Where did this suddenly come from. If you want to write a section about the effect of one point and two point measurements of the storage term this should be given a section of its own. Please clarify in the method section how the one point and two point storage terms were estimated. This has nothing to do with partitioning.

Section 3.2 What about the partitioned GPP and ecosystem respiration data? Why did you not show the diurnal cycle of them? Additionally, what was the environmental variables controlling the diurnal dynamics. You set out in the introduction to investigate which environmental variables that affect the diurnal, seasonal and interannual dynamics, but there is no proper description of what controls the diurnal dynamics.

Section 3.3

L235 Why did you analyse using monthly averages? Is there any reason for not using daily averages? A lot of dynamics can be hidden in case you average like that.

How did you test for all these things that you claim? You state that rainfall and low VPD causes the seasonal dynamics, but there is absolutely no statistical tests done to show that these variables are determining the fluxes? A large part of the section is rather about interannual dynamics than about seasonal variation.

L266 from which date to which date?

Why is NDVI included as an own section and not just incorporated in the other sections as an explanatory variable giving the phenology of the vegetation?

L274 The Merbold study is rather in the spatial domain than in the temporal domain, and it is comparing sites from tropical rain forest to semi-arid savanna ecosystems so it is not strange that they see a spatial relation to rainfall.

L275 Which results show this?

What about interannual variation in respiration?

L285 Demokeya has approximately 7% tree cover, i.e. about half of Welgegund. This is not a similar canopy cover.

L287 Why is it that NDVI over and underestimates at different parts of the season?

L309 The strong grazing pressure cannot explain the difference as there is a very strong grazing pressure at the Dahra field site as well. Please see (Tagesson et al., 2016b). How come that the missing data of Tagesson et al can explain the difference between Dahra and Welgegund? The uncertainty estimates of Tagesson et al indicate that the missing data should not be a reason for huge uncertainty in the annual budgets?

Have you tried to make any uncertainty estimate of the annual budgets?

Table 1, please explain what the abbreviations under Species are? Example what does P.J.H.Hurter mean?

Table 3 and 4, please give exact sampling dates instead of column 1-4, it seems like Table 3 and 4 can be combined.

Figure 1, please include error bars.

Figure 2. Why did you use a different point for the NDVI comparison? If this is the case, then it is more important to show the MODIS pixel surrounding that point. In case you want to present the results for all different vegetation types, please show the transects used for the vegetation samplings.

Figure 3 please include GPP and respiration

Figure 4 Why was the data binned and what was the size of the bins? When was this data used?

Figure 5 how come that soil moisture is up to 20% during the dry season, this is very high.

Figure 6 I like this figure, except why monthly sum per hour, it is just a confusing unit. Why not hourly average, it would make more sense?

Figure 7 why not include all months? Again why hourly per month. February with 28 days will be different than January just because of # of days in the month.

**References**

Tagesson, T. et al., 2016a. Spatiotemporal variability in carbon exchange fluxes across the Sahel Agric. For. Meteorol., 226–227: 108-118.

Tagesson, T. et al., 2016b. Very high carbon exchange fluxes for a grazed semi-arid savanna ecosystem in West Africa. Danish Journal of Geography: accepted.

---

## Author Comment (AC1) · 26 Oct 2016

**Final Authors Response**

We thank both reviewers for their extensive and insightful feedback.

**Anonymous Reviewer  #1**

The paper by Räsänen et al. explores carbon dioxide fluxes measured with the eddy covariance method for three years at a grazed savanna grassland in Welgegund, South Africa. The material is appropriate for a scientific study and the data obtained appear to be high-quality. It is relevant for many African ecosystems to focus on CO2 fluxes response to environmental drivers in order to better predict fluxes patterns in the context of climate change. Therefore, the work is interesting and worthy of publication in Biogeosciences Journal because of the lack of knowledge regarding the carbon cycle for Africa continent. However, I have a number of issues with the paper which lead me to suggest that it requires major revisions before it becomes acceptable for publication in BG.

**General Comments:**

**1)** Firstly, while the study site is located on a savanna grassland which is grazed by cattle and sheep, authors did not provide any information on the average stocking rate and the management of the site during the studied period. Is the site grazed intensively or not? What was the stocking rate? How the grassland was managed?

**Authors' response:** Unfortunately more detailed measurements of cattle respiration were not available for this study. We added a paragraph to the site description about the farm management which is a typical commercial farm in South Africa.

*"The measurement site is located at a commercial farm which has about 1300 head of cattle which varies ± 300 depending on the year. During a wet year there are more animals than during a dry year. The cattle are grazing on an area of approximately 6000 ha, which consists of natural grazing (e.g. at the measurement site), planted grazing and maize/sunflower fields that are grazed after harvesting. This form of farming is considered large-scale commercial farming. Due to the semi-arid climate, the carrying capacity of the grazing fields tends to be low and thus the grazing area is large. The farmers cannot keep track of the grazing patterns, but they do move the cattle around to optimize grazing and protect the field against overgrazing."*

What is the slope of the field? At the measurement height what is the fetch? Was the fetch adequate to characterize the carbon dioxide and water vapor fluxes of the vegetation type? These are important for understanding and interpreting the results.

**Authors' response:** The measurement site is surrounded by flat homogeneous thornveld. As shown by the footprint climatology and the land-use map, the fetch is adequate for measuring fluxes over this vegetation type.

**2)** It is also well known (see references below) that grazing affects a range of ecological and biogeochemical processes and properties, including plan community composition, soil physical

properties, soil C and nitrogen content and the magnitude of C and carbon dioxide exchanges which in turn influence soil organic carbon storage. This study could have been more attractive if the impact of grazing on carbon dioxide exchange had been investigated. This probably would help to better assess for example the relation between the total ecosystem respiration and environmental drivers.

**Authors' response:** It is true that there is heavy grazing within the measurement footprint and that affects a range of processes. Unfortunately, more detailed study of the grazing effects was not possible here.

**3)** Authors used the Kaimal cospectra in the computation of the correction factors that are used to correct the high frequency losses (L129 – 130). However, recent studies (Mamadou et al., 2016) showed that Kaimal cospectra can be significantly different from sensible heat cospectra, and the high-frequency loss correction for CO2 using these different cospectra resulted in the large difference in CO2 flux calculations, i.e., using Kaimal cospectra can result in an overestimation of CO2 fluxes even if the site could not be considered as difficult (i.e., fairly flat, homogeneous, low vegetation, sufficient measurement height). Especially, at their studied site, authors found that the choice of Kaimal rather than sensible heat cospectra reversed the annual carbon balance from being a net C sink to being a weak C source. Did the authors verify if their kaimal cospectra differ or not from sensible heat cospectra before chosen them as idealized cospectra?

**Authors' response:**
The use of the so-called Kaimal cospectra is a common practice in eddy covariance studies (e.g. Aubinet et al.: Eddy Covariance, A Practical Guide to Measurement and Data Analysis, 2012). We followed this practice and used Kaimal cospectra with a system-specific transfer function for correcting for the flux losses in question. Mamadou et al. (2016) have very recently (unavailable at the time of writing of our paper) published a paper that indicates that, for an unspecified reason, the local cospectra at their site differ from the generic Kaimal cospectra. While this is an interesting observation that deserves attention in the future, it is not obvious that the implications of the potential differences would be as significant at other sites as their results may imply. It should be noted that the cut-off frequency of their measurement system was 0.37 Hz, while we were able to resolve much higher frequencies (half-power frequency 1.6 Hz). Thus our flux loss corrections are much smaller than those applied by Mamadou et al. (2016), 5% on average and <10% in 98% of the data. If we assume that our correction coefficients are uncertain by a factor similar to that estimated by Mamadou et al. (2016), the flux uncertainty resulting from these small correction coefficients would be minor. Therefore, in the present study, we do not pursue the issue of spectral corrections any further; however, we did add the uncertainty related to flux loss corrections in our uncertainty estimate for the annual $CO_2$ balance.

**4)** Most of results presented in the section 3.4 are too much qualitative, superficial and descriptive and should be supplemented with additional statistical analyses in order to provide more quantitative rigor.

**Authors' response:** The data covered only three years and thus a statistical analysis of annual averages is not very meaningful. To improve the presentation, the differences between the years were analysed from monthly data, including statistical analysis.

**5)** The uncertainties associated to the annual carbon dioxide balance estimation are not evaluated. This remains a great lack for the study. The authors also clearly mentioned in their introduction that environmental drivers for the inter-annual variation in NEE are poorly understood. Unfortunately no progress regarding this point has been made within the present study.

**Authors' response:** New subsection about error estimation was added to the methods section and the uncertainty of annual carbon dioxide balance was estimated.

**Specific comments**

**L18-19:** What about the dependence, at monthly scale, of the nighttime respiration on soil moisture or soil temperature?

**Authors' response:** For the gap-filled monthly sum of the night-time respiration, the relation with soil temperature is exponential. For the soil moisture the relation is not clear.

**L24-25:** by increasing autotrophic respiration?

**Authors' response:** Probably, but we cannot infer that from total ecosystem respiration.

**L32:** The seasonal cycle of what? Please clarify.

**Authors' response:** Rephrased. "The savanna ecosystems are generally characterized by alternating wet and dry seasons, during the latter of which wildfires can occur."

**L32-33:** The alternation of "wet and dry seasons" cannot in my view be generalized for the "whole Africa". In other regions of Africa, the dry and wet seasons are separated for example by two transitional seasons...

**Authors' response:** Added a sentence about transtional seasons.

**L67,** in site description section: Please, give values of the roughness length, zero-displacement height and site's slope.

**Authors' response:** The median of the aerodynamic roughness length was estimated to be 0.42 m assuming a low zero-plane displacement height.

**L102-103:** Specify the sampling rate of the meteorological variables.

**Authors' response:** The meteorological variables were sampled every minute and 15 min averages were recorded.

**L113:** Specify the type of the gas analyzer.

**Authors' response:** Specified.

**L115-118:** What are the characteristics of the sampling tube (inner diameter etc.), the pump and the gas used for the zero and span?

**Authors' response:** Corrected. *"The material of the inlet tube (ID 4mm, OD 6mm) was PTFE, and the pump was Dürr A-062 E1. The gas analyzer was calibrated every month with a high-accuracy $CO_2$ span gas (378 ppm verified by the Cape Point GAW station), and Afrox instrument grade synthetic air with $CO_2$ < 0.5 ppm was continuously used as a reference gas."*

**L127:** Give an indication of the magnitude of low frequency correction factors.

**Authors' response:** We added statististics on the magnitude of spectral corrections to the text according to the data presented above.

**L129-130:** Provide an illustration of kaimal and the sensible heat cospectra according atmospheric stability to attest that both cospectra match.

**Authors' response:** Since the flux loss corrections required for our data are small, irrespective of the reference cospectrum adopted, we did not study the spectral characteristics further here; however, we included a related uncertainty estimate, as explained above.

**L133:** Replace the calculated fluxes by "the corrected fluxes".

**Authors' response:** Corrected.

**L133:** What was the fraction of data excluded this way?

**Authors' response:** u* filtering excluded 18 % of the data.

**L133-136:** Do you only use u* filtering criteria to discard bad data? if Yes, explain why.

**Authors' response:** *"In addition, $CO_2$ fluxes were filtered by setting an acceptable range for average $CO_2$ concentration (300–500 ppm), Licor pressure (50–120 kPa) and $CO_2$ concentration variance (0–10 $ppm^2$), which resulted in a 3 % loss of flux data in total."*

**L181:** Complete "air" with temperature.

**Authors' response:** Corrected.

**L182:** You never indicated how water vapor data have been treated. What is the cut-off frequency for H2O fluxes? How these data have been corrected for low and high frequency losses? Which criteria have been used for the filtering of bad data?
**L183:** Explain how high evapotranspiration rate were due to higher precipitation and transpiration rate during the rainy season? What about soil evaporation?

**Authors' response:** The evapotranspiration data were excluded from the present analysis.

**L206:** air or soil temperature?

**Authors' response:** Soil temperature

**L211- 215:** The low (high) values of the correlation coefficients cannot only be used to attest the robustness of dependences. These must be accompanied with the p-values.

**Authors' response:** The method comparison was removed and the Lloyd and Taylor (1994) model was used for fitting the night-time respiration data. The modelled respiration rates agreed well with the measured respiration ($R^2$=0.56, p-value < 0.01).

**L225- 226:** showed how?

**Authors' response:** The peak carbon uptake can be seen from the darkest pixels in Figure 6. Peak radiation was checked from data (data not shown).

**L232:** I cannot get this conclusion...

**Authors' response:** Added supporting figure to the supplement. Relationship between bin averaged VPD and daytime NEE was plotted.

**L223–L233:** Why is there so much interpretation in the results?

**Authors' response:** The text in this section was rephrased and an analysis of diurnal cycle of GPP, respiration and VPD was added to the section.

**L301:** most or must?
**Authors' response:** most

**L305-310:** I am afraid that because of the difference of their climate, the Dahra site and cannot be easily compared to the Welgegund site. You should mention this in your discussion.

**Authors' response:** Removed the sentence and added "The large difference in the carbon balance is due to much larger carbon uptake at Dahra during the rainy seasons which might be explained by moderately dense $C_4$ ground vegetation and high soil nutrient availability."

**L315, L317:** Write Nalohou not Nolohou. . .

**Authors' response:** Corrected.

**L475:** Figure 1 and also in the title: "air" or "soil" temperature?

**Authors' response:** Corrected.

**L500:** Figure 3: Is it necessary to show evapotranspiration curve?

**Authors' response:** The evapotranspiration curve was removed.

**L522:** Figure 4: bin averaged for how many data?

**Authors' response:** The figure shows 2814 values and each bin contains at least 100 values.

---

## Author Comment (AC2) · 26 Oct 2016

**Final Authors Response**

We thank both reviewers for their extensive and insightful feedback.

**Anonymous Reviewer #2**

Major concerns:

Generally, the language needs to be improved and the manuscript must be much more concise. This goes for all sections. A large section of the results is about testing different partitioning methods, but it is not included as an aim in the introduction. In case an aim of the study is to investigate different partitioning methods, please clarify this already in the introduction. Otherwise, I would recommend using the one that works the best. I fully understand why the night-time methods are not working well, since it has generally been seen that at a diurnal time scale respiration is not strongly linked to temperature for savanna ecosystems. But I cannot understand why the daytime method is working so poorly. Are you sure that you fitted the equation correctly? After putting a lot of effort into partitioning of the data, the partitioned data is not even shown except in the format of monthly averages. Please, show the partitioned data as well.

**Authors' response:** The testing of partitioning methods was not a major goal of this study. The method comparison has been removed and the Lloyd and Taylor (1994) model was used for fitting the night-time respiration data. The manuscript was made more concise and the language was improved.

Why did you use monthly averages in the investigation of seasonal variation? I cannot see any reason for not using daily sums or averages. By using monthly estimates you hide a lot of the variability and it is more difficult to see the relationship to the environmental variables.

**Authors' response:** The kind of analysis we had done would be very difficult to do at daily time scale because the daily data is noisier and the data covered three years.

There is no statistical testing of relationships of the fluxes to the environmental variables. It is just stated that high flux values can be explained by some variables. But you do not explain how you have tested for this.

**Authors' response:** Results of linear regression analysis between NEE and environmental drivers were added to the section 3.3.

**Specific comments:**

**L19-20.** There is a contradiction in this sentence: are the balances yearly or are they for the growing season,please rephrase.

**Authors' response:** Corrected.

**L23-24** Please clarify in the abstract why: This study underlines the difficulty in establishing a functional relation between the total ecosystem respiration and the environmental drivers in savanna ecosystems.

**Authors' response:** Sentence removed.

**L24-25** There must be something that explains the inter-annual variability, even though it might be that you have not seen any explanations in your data sets.

**Authors' response:** Sentence removed.

**L32** reference for the 20% of global area please.

**Authors' response:** Same reference.

**L35**. Please rephrase, it sounds like the humans are grazing.

**Authors' response:** Corrected.

**L42** reference please

**Authors' response:** Added.

**L45** . instead of ,

**Authors' response:** Corrected.

**L51** This is not correct: Tagesson, Ago and Quansah is all sites affected by either grazing or agriculture, there are also EC towers in Wankama Falls, Agofou, and Demokeya, which are all affected by grazing or agriculture; see (Tagesson et al., 2016a)

**Authors' response:** Corrected.

**L55** please include (Tagesson et al., 2016a) that investigated annual budgets for 6 different sites across the Sahel.

**Authors' response:** Included.

**L61** I do not understand why you use NDVI as a proxy for GPP. You have EC measurements, why not use them directly? Please clarify.

**Authors' response:** This was done to assess the long-term (in this case 12 years) productivity at the measurement site.

**L70** Do you have any data on number of sheep and cattle?

**Authors' response:** We added a paragraph to the site description about the farm management which is a typical commercial farm in South Africa.

*"The measurement site is located at a commercial farm which has about 1300 head of cattle which varies +- 300 depending on the year. During a wet year there are more animals than during a dry year. The cattle are grazing on approximately 6000 hectars, which consists of natural grazing (e.g. at the measurement site), planted grazing and maize/sunflower fields that are grazed after harvesting. This form of farming is considered large-scale commercial farming. Due to the semi-arid climate, the carrying capacity of the grazing fields tends to be low and thus the grazing area is large. The farmers cannot keep track on the grazing patterns but they do move the cattle around to optimize grazing and protect the field against overgrazing."*

**L86** please give exact sampling dates.

**Authors' response:** Exact sampling dates were added to the Table 3.

**L91** What do you mean, did you count all plants inside the 100m2 plot or did you identify all species?

**Authors' response:** *"All of the plant species were counted and identified up to species level,and their major growth form was recorded."*

**L101** Are not all these measurements relevant to Ecosystem dynamics? Please use a different word than ecosystem dynamics.

**Authors' response:** Carbon cycle dynamics.

**L103** At 2 and 8 m height or at several heights between 2 and 8 m? What was the height of the tipping buckets?

**Authors' response:** Rephrased. Tipping bucket was installed at 1.5 m height.

**L102** What sensors were used for the meteorological measurements?

**Authors' response:** Corrected.

*"The meteorological measurements included air temperature (Rotronic MP 101A) and pressure (Vaisala PTB100B), wind speed (Vector A101ML) and direction (Vector A200P/L), relative humidity and temperature gradient (Vaisala PT-100) between two points (2 m and 8 m height)"*

**L108** What sensor? what did it measure?

**Authors' response:** Soil moisture sensor at 5 cm depth.

**L115** Why 20 m when height of the sensor was only 9m, should be possible to have a much shorter tube than this. What sort of tube did you use, inner diameter? No filter between the IRGA and the incoming air?

**Authors' response:** The gas analyzer is located inside a trailer that is some distance away from the measurement mast. The gas sampling tube was PTFE (ID 4mm, OD 6mm) and it had a filter.

What was the separation length between the inlet tube and the anemometer?

**Authors' response:** 20cm

**L137-140** and Figure 2. The footprint is never this uniform for different wind directions, if you have estimated the footprint for each 30 min period; it would be easy to show an average for the different wind directions.

**Authors' response:** Added contour of the mean 80% cumulative footprint.

**L140** If the footprint is homogeneous thornveld, why do you report vegetation sampling for all other vegetation types? Looking at figure 2 with the footprint, is seems like the only vegetation cover which is affecting the EC measurements are the thornveld. If you want to present all the other data as well, I think you should you must incorporate a reason for this in the introduction, and a link to the EC data.

**Authors' response:** The reviewer is correct: thornveld clearly dominates the flux footprint and that is the vegetation type we focus on. For clarity, Section 2.2 on vegetation sampling and the tabulated soil data, which are included to characterize a larger area, were moved to Supplement.

**L145-151** Please give equations for the all partitioning models.

**Authors' response:** The method comparison has been removed and the Lloyd and Taylor (1994) model was used for fitting the night-time respiration data.

**L163** Why in two steps? Why give E0 an annual value and not using the moving window?

**Authors' response:** This makes the fitting procedure more robust.

**L166** What is Fp? Why fitting this at all? Why not using the light response function that was used in the "daytime method"?

**Authors' response:** Fp is GPP. The fitting was done following the method by Kutsch et al. (2008).

**L170** This is probably a good choice, but why not always use 1 September to 31 August or something similar? Why only estimating the growing seasons?

**Authors' response:** Rephrased the sentence. The analysis covered the whole years.

**L174** This is incorrect MCD43A4 is not an NDVI product, it is a BRDF product, please rephrase. Why using monthly averages, when data is available as an 8 day product? It is not clear if you extracted the values of one single pixel, or did you use an average of several pixels?

**Authors' response:** Rephrased. Monthly average value of one 500 m pixel was used.

**L176** What do you mean by that NDVI is better to use than the LAI and GPP product for vegetation structure? First, you are not studying vegetation structure, you are studying fluxes. Different products are good for different things. I agree that NDVI is a useable parameter, but it is not a real value like LAI and GPP. You cannot claim that it is better to use than these other parameters for studying vegetation dynamics. It is so far very unclear what you are going to use these data for. Please clarify. I would state that NDVI is a proxy for vegetation phenology.

**Authors' response:** Removed the last sentence. The NDVI was used to study long-term productivity at the measurement site.

**L180** please give sum of rain

**Authors' response:** Corrected.

**L192** In figure 4, it does not look like a linear increase until the saturation level. It is rather asymptotic

**Authors' response:** Removed "linear".

**L197** How could the parameters get unrealistically high? In case data looks like in Figure 4, parameters should be fine. In case the relationship is very linear, the saturated GPP level gets unrealistically high, but this does not really matter for the partitioning as long as the equation is well fitted to the data. As the manuscript is written now, it seems like you want to test several different partitioning methods, if this is the case then you must show how all these different partitioning methods differ in their output. In the results section, you do not show the output of the different methods at all.

**Authors' response:** The method comparison has been removed.

**L209** I would not say that it was fitted successfully in case the R2 value is 0.11.

**Authors' response:** This result was removed.

**L210-215**, in case you want to make a proper comparison of the different partitioning methods, you should give statistics for all methods. Please also show a figure with modelled versus measured values. In order to make a proper comparison you must separate a part of the data set to be used for the model parameterisation and one part for the model evaluation. A suggestion would be to use a bootstrapping simulation methodology.

**Authors' response:** The method comparison has been removed and the Lloyd and Taylor (1994) model was used for fitting the night-time respiration data. The modelled respiration rates correlated with the measured respiration ($R^2$=0.56, p-value < 0.01).

**L216** Where did this suddenly come from. If you want to write a section about the effect of one point and two point measurements of the storage term this should be given a section of its own. Please clarify in the method section how the one point and two point storage terms were estimated. This has nothing to do with partitioning.

**Authors' response:** This paragraph was moved to the section 3.4.

**Section 3.2**

What about the partitioned GPP and ecosystem respiration data? Why did you not show the diurnal cycle of them? Additionally, what was the environmental variables controlling the diurnal dynamics. You set out in the introduction to investigate which environmental variables that affect the diurnal, seasonal and interannual dynamics, but there is no proper description of what controls the diurnal dynamics.

**Authors' response:** The diurnal cycle of GPP, ecosystem respiration and VPD was added to the section 3.2 and their relation to the diurnal cycle of NEE was analyzed.

**Section 3.3**

**L235** Why did you analyse using monthly averages? Is there any reason for not using daily averages? A lot of dynamics can be hidden in case you average like that. How did you test for all these things that you claim? You state that rainfall and low VPD causes the seasonal dynamics, but there is absolutely no statistical tests done to show that these variables are determining the fluxes? A large part of the section is rather about interannual dynamics than about seasonal variation.

**Authors' response:** The kind of analysis would have been very difficult to do at daily time scale because the daily data is noisier and the data covered for three years. Linear regression analysis between NEE and the environmental variables was added to the section.

**L266** from which date to which date?

**Authors' response:** from $1^{st}$ of September to $31^{st}$ of August

Why is NDVI included as an own section and not just incorporated in the other sections as an explanatory variable giving the phenology of the vegetation?

**Authors' response:** It is used to assess the long-term productivity at the measurement site. *"Based on the NDVI data, the year 2010-2011 represents a common pattern at this site, whereas the years 2011-2012 and 2012-2013 have lower NDVI peak values."*

**L274** The Merbold study is rather in the spatial domain than in the temporal domain, and it is comparing sites from tropical rain forest to semi-arid savanna ecosystems so it is not strange that they see a spatial relation to rainfall.

**Authors' response:** Removed the sentence.

**L275** Which results show this?

**Authors' response:** The sentence was removed.

What about interannual variation in respiration?

**Authors' response:** The relation between the annual respiration and environmental drivers was not clear.

**L285** Demokeya has approximately 7% tree cover, i.e. about half of Welgegund. This is not a similar canopy cover.

**Authors' response:** Corrected.

**L287** Why is it that NDVI over and underestimates at different parts of the season?

**Authors' response:** The sentence was removed.

**L309** The strong grazing pressure cannot explain the difference as there is a very strong grazing pressure at the Dahra field site as well. Please see (Tagesson et al., 2016b). How come that the missing data of Tagesson et al can explain the difference between Dahra and Welgegund? The uncertainty estimates of Tagesson et al indicate that the missing data should not be a reason for huge uncertainty in the annual budgets?

**Authors' response:** Removed the sentence and added "The large difference in the carbon balance is due to much larger carbon uptake at Dahra during the rainy seasons which might be explained by moderately dense $C_4$ ground vegetation and high soil nutrient availability."

Have you tried to make any uncertainty estimate of the annual budgets?

**Authors' response:** New subsection about error estimation was added to the methods section and the uncertainty of annual carbon dioxide balance was estimated.

**Table 1**, please explain what the abbreviations under Species are? Example what does P.J.H.Hurter mean?

**Authors' response:** Specified. The plant species names include the name of the publishing author.

**Table 3 and 4**, please give exact sampling dates instead of column 1-4, it seems like Table 3 and 4 can be combined.

**Authors' response:** Exact sampling dates were added and Table 3 and 4 were combined.

**Figure 1**, please include error bars.

**Authors' response:** Corrected.

**Figure 2**. Why did you use a different point for the NDVI comparison? If this is the case, then it is more important to show the MODIS pixel surrounding that point. In case you want to present the results for all different vegetation types, please show the transects used for the vegetation samplings.

**Authors' response:** The NDVI for the measurement site was calculated at the 500 m pixel indicated by the red square in Figure 2. The NDVI comparison point in Figure 2 indicates the moist sandy grassland area which is not grazed. The NDVI signal between these two areas were compared.

*"From September 2001 to August 2013, the yearly maximum values of the measurement site NDVI were 0.02 units smaller on average than a nearby moist sandy grassland area which is not grazed (land-use class 6 in Figure 2). This difference is most probably due to heavy grazing at the measurement site."*

**Figure 3** please include GPP and respiration

**Authors' response:** Time series of daily sums of NEE, GPP and respiration was added to the Supplement (Figure S5).

**Figure 4** Why was the data binned and what was the size of the bins? When was this data used?

**Authors' response:** To better visualize the light response curves during the wet and dry seasons. The figure shows 2814 values and each bin has about 100 values.

**Figure 5** how come that soil moisture is up to 20% during the dry season, this is very high.

**Authors' response:** There was a 22 mm precipitation event in 7[th] of June 2011. During that time the soil moisture peaked at 18 %.

**Figure 6** I like this figure, except why monthly sum per hour, it is just a confusing unit. Why not hourl average, it would make more sense?

**Authors' response:** Correct to mean NEE for each hour of day.

**Figure 7** why not include all months? Again why hourly per month. February with 28 days will be different than January just because of # of days in the month.

**Authors' response:** All months were included. Changed units to mean diurnal cycle of NEE.

**References**

Kutsch, W. L., Hanan, N., Scholes, B., McHugh, I., Kubheka, W., Eckhardt, H. and Williams, C.: Response of carbon fluxes to water relations in a savanna ecosystem in South Africa, Biogeosciences, 5(6), 1797–1808, doi:10.5194/bg-5-1797-2008, 2008.

---

## Referee Report (RR1)

**Comments to the authors of "Carbon balance of a grazed savanna grassland ecosystem in South Africa"**

I would like to congratulate the authors to a truly improved manuscript. It is very much better than the previous version; the text is better structured and the message is much clearer. I am also happy that an uncertainty analysis has been conducted; this is far too often overseen in papers presenting annual $CO_2$ flux budgets. Well done. However, I still have some questions and comments remaining.

**Comments:**

L81 What do you mean by low zero displacement height? Please give a value and explain why it was set to this.

L163-L164. Please include an explanation to why $E_0$ was estimated on an annual basis whereas $R_b$ was estimated on an 6 day basis. Does it really make physical sense that $E_0$ is not changed over the growing season?

L189 How do you know that ustar 0.2 is the optimal ustar threshold? It is quite high in relation to most studies. For a method to detmine the optimal threshold see Lund et al. (2007).

L186. Are you certain that your ustar filtering is a random error? More data is usually filtered during morning and during night time, indicating that it is a systematic error rather than a random error. Please see Moncrieff et al. (1996). You do not have to do the full analyis of investigating which errors are random and systematic as in Moncrieff; it is a quite big job. But if you want to keep ustar filtering as a random error, please explain why.

It is very good that you did an uncertainty analysis, however we all know that a true quantification of all uncertainties related to the eddy covariance method is very hard to do. What about all instrumentation errors, uncertainities related to all different kind of preproccessing choices, errors related to the WPL correction, etc. I would just like you to put in an humble sentence describing that this is an estimate including some important uncertainities, but not all, affecting the annual $CO_2$ flux budgets.

L199 the systematic errors should be summed without being taken in quadrate; it is only the random errors that should be taken in quadrate. Please see Moncrieff et al. (1996).

L201 I am trully sorry, I was slightly sloppy in my previous review. MCD43A2 is not a BRDF product, it is a NBAR (nadir BRDF adjusted reflectance) product.

Fig S1 Please also include an explanation to what the NDVI comparison point is in the Figure caption. When only looking at the figure I thought this was where you took your NDVI value from. Later when reading the text I realised, that it was the point without grazing.

Fig S2 Please do like in Fig 3, include all data as well as the binned data.

L217. I still do not understand why you did not use the light response function instead of the Lloyd Talyor equation when partitioning NEE into GPP and Reco. You did not see any clear relationship between night time respiration and temperature, would it then not be better to use a relationship where you see a strong correlation?

L240-L243 Why don't you combine Fig 6, Fig S3 and Fig S4, it would make it easier to see the relationship between VPD and the fluxes. Why is the Fig S3 and Fig S4 in the supplementary material, they seem to be quite important for the results?

Section 3.3 Should probably be called intra-annual, since it is investigating sesonal dynamics.

L255, I still do not understand why you using monthly data in your analysis. Your dailys sums are beautiful and not too noisy (Fig S5). The variability seen in S5 is the variability you want to explain, but that variability disappears when taking the monthly averages. If you necessarily want to keep the monthly averages you should give an explanation for this in the manuscript.

L321-L324 Please rephrase this sentence. It was really hard to understand what you meant.

L364 Which previous studies? I think that it is mainly in humid savanna ecosystems where there is a clear relationship between ecosystem respiration and temperature, whereas dry savanna ecosystems generally have not seen a clear relationship.

Table 1. I am sorry, I still do not understand what the publishing author is? Is it the first person finding this species? Please clarify this and the acronyms used in the table.

Fig 3 and Fig S2. Please explain in the figure caption how the bins were defined, between 0-150 PAR, 150-450 PAR etc. Or? They are not defined to have 100 values per bin, then there would be 28 points.

Fig 7, legend 4. What is yearly sum?

**References:**

Lund M et al. (2007) Annual CO2 balance of a temperate bog. *Tellus Series B.* 59B, 804–811

Moncrieff, J. B., Malhi, Y., & Leuning, R. (1996). The propagation of errors in long-term measurements of land-atmosphere fluxes of carbon and water. *Global Change Biology, 2*, 231–240. doi:http://dx.doi.org/10.1111/j.1365-2486.1996.tb00075.x

---

## Author Response (AR2)

**Final Authors Response**

We thank the anonymous reviewer #2 for helpful comments.

**Anonymous Reviewer  #2**

I would like to congratulate the authors to a truly improved manuscript. It is very much better than the previous version; the text is better structured and the message is much clearer. I am also happy that an uncertainty analysis has been conducted; this is far too often overseen in papers presenting annual CO2 flux budgets. Well done. However, I still have some questions and comments remaining.

**Comments:**

**L81** What do you mean by low zero displacement height? Please give a value and explain why it was set to this.

**Authors' response:** The zero displacement height was set to zero because the grass is heavily grazed and thus it is usually not taller than about 5cm. The text was changed accordingly.

**L163-L164.** Please include an explanation to why E0 was estimated on an annual basis whereas Rb was estimated on an 6 day basis. Does it really make physical sense that E0 is not changed over the growing season?

**Authors' response:** The E0 parameter was tested to vary freely together with Rb parameter in which case most of the E0 values were below zero.  On the other hand, in the two step fitting the Rb parameter had clear seasonal cycle. A similar approach was used in a flux partition comparison study by (Lasslop et al., 2010).

**L189** How do you know that ustar 0.2 is the optimal ustar threshold? It is quite high in relation to most studies. For a method to detmine the optimal threshold see Lund et al. (2007).

**Authors' response:** The ustar threshold was determined using the same approach as in Lund et al. (2007). Added sentence describing the method at line 140.

**L186.** Are you certain that your ustar filtering is a random error? More data is usually filtered during morning and during night time, indicating that it is a systematic error rather than a random error. Please see Moncrieff et al. (1996). You do not have to do the full analyis of investigating which errors are random and systematic as in Moncrieff; it is a quite big job. But if you want to keep ustar filtering as a random error, please explain why.
It is very good that you did an uncertainty analysis, however we all know that a true quantification of all uncertainties related to the eddy covariance method is very hard to do. What about all instrumentation errors, uncertainities related to all different kind of preproccessing choices, errors related to the WPL correction, etc. I would just like you to put in an humble sentence describing that this is an estimate including some important uncertainities, but not all, affecting the annual CO2 flux budgets.

**Authors' response:** It is true that the ustar filtering is selectively systematic error. This was corrected to the text. The beginning of the Section 2.5 now reads as follows *"The uncertainty in the annual CO2 budget was estimated by considering the most significant, albeit admittedly not all possible error sources, for both random and systematic errors."*

**L199** the systematic errors should be summed without being taken in quadrate; it is only the random errors that should be taken in quadrate. Please see Moncrieff et al. (1996).

**Authors' response:** We disagree with this point because both systematic and random errors should be added in quadrature. This is also demonstrated in the discussion section of the Moncrieff et al. (1996) paper. All errors regardless of their type should be added in quadrature if they are independent errors i.e. uncorrelated.

**L201** I am trully sorry, I was slightly sloppy in my previous review. MCD43A2 is not a BRDF product, it is a NBAR (nadir BRDF adjusted reflectance) product.

**Authors' response:** Corrected.

**Fig S1** Please also include an explanation to what the NDVI comparison point is in the Figure caption. When only looking at the figure I thought this was where you took your NDVI value from. Later when reading the text I realised, that it was the point without grazing.

**Authors' response:** Corrected.

**Fig S2** Please do like in Fig 3, include all data as well as the binned data.

**Authors' response:** Corrected.

**L217.** I still do not understand why you did not use the light response function instead of the Lloyd Talyor equation when partitioning NEE into GPP and Reco. You did not see any clear relationship between night time respiration and temperature, would it then not be better to use a relationship where you see a strong correlation?

**Authors' response:** The light response functions were tested extensively by our own implementation of Lasslop et al., (2010) (Lasslop et al., 2010) and by the Dynamic INtegrated Gapfilling and partitioning for Ozfluz (DINGO) implementation (Beringer, (2014)). The majority of the fit parameters were not in sensible range and forcing them to certain range as was done in Lasslop et al., (2010) left very few fitted parameter values. Partitioning was also tested without forcing the parameters to any range which resulted in negative daytime respiration for the dry season months June, July and August. The light response functions are challenging to fit as pointed out by Aubinet et al., (2012) and with our data it was not possible to get sensible results using this approach.

**L240-L243** Why don't you combine Fig 6, Fig S3 and Fig S4, it would make it easier to see the relationship between VPD and the fluxes. Why is the Fig S3 and Fig S4 in the supplementary material, they seem to be quite important for the results?

**Authors' response:** The combination of carbon flux and VPD plots was tested but there was too much overlap with the plot lines. The Figures S3 and S4 were combined with Fig 6 and the VPD figure was added to the article.

**Section 3.3** Should probably be called intra-annual, since it is investigating sesonal dynamics.

**Authors' response:** Corrected.

**L255**, I still do not understand why you using monthly data in your analysis. Your dailys sums are beautiful and not too noisy (Fig S5). The variability seen in S5 is the variability you want to explain, but that variability disappears when taking the monthly averages. If you necessarily want to keep the monthly averages you should give an explanation for this in the manuscript.

**Authors' response:** It is true that the monthly values did hide the early wet season transition point after which the ecosystem is mostly taking up carbon. These dates were added to the analysis. An explanation for the monthly scale analysis was added to the beginning of Sec. 3.3. *"This analysis was based on monthly data in order to facilitate comparison between the years, which would not be possible with daily data due to the large scatter in carbon fluxes and environmental variables."*

**L321-L324** Please rephrase this sentence. It was really hard to understand what you meant.

**Authors' response:** Rephrased. The sentence now reads as follows *"Within our three year flux data period, the peak NDVI was related to the peak GPP in year 2010-2011 and in year 2011-2012 but it did not capture the rainy season peak in GPP in 2012-2013."*

**L364** Which previous studies? I think that it is mainly in humid savanna ecosystems where there is a clear relationship between ecosystem respiration and temperature, whereas dry savanna ecosystems generally have not seen a clear relationship.

**Authors' response:** There was a mistake in this sentence. The sentence now reads as follows *"Similar results were observed at a West-African dry savanna, where none of the environmental variables could explain the half-hourly night-time respiration measurements (Tagesson et al., 2015)."*

**Table 1.** I am sorry, I still do not understand what the publishing author is? Is it the first person finding this species? Please clarify this and the acronyms used in the table.

**Authors' response:** The species names in the table are taken from the website www.theplantlist.org which is an effort provide standard accepted names for all plant species. The figure caption now reads as follows *"Dominant plant species within the flux footprint. The plant species name is written in italics whereas the roman text refers to the author. Some author names are abbreviated."*

**Fig 3 and Fig S2.** Please explain in the figure caption how the bins were defined, between 0-150 PAR, 150-450 PAR etc. Or? They are not defined to have 100 values per bin, then there would be 28 points.

**Authors' response:** The figures were made again with equal sized bins. The captions now read the exact amount of data in each bin.

**Fig 7**, legend 4. What is yearly sum?

**Authors' response:** Yearly precipitation sum. Corrected the legend.

[revised manuscript text omitted]